# Signal-strapping as a protein-sequence search method for the discovery of metalloproteins

João Paulo L. Franco Cairo ◉[1], Thamy L. R. Corrêa[1], Wendy A. Offen ◉[1], Alison K. Nairn ◉[1], Julia Walton[1], Sean T. Sweeney ◉[2] ✉, Gideon J. Davies ◉[1] ✉ & Paul H. Walton ◉[1] ✉

Metalloprotein discovery is often made *post hoc*, in which activity studies following protein isolation reveal a metal-ion dependence. Herein we take a different approach to finding metalloproteins, by building on the discovery of copper-containing *lytic polysaccharide monooxygenases* (LPMOs), which include an N-terminal histidine as part of their sequence. This residue acts as a natural chelator for transition metal ions, irrespective of the structure of the protein. We report the method of *signal strapping*, where sequences of N-terminal signal peptides artificially appended with a histidine residue at their C-terminus are used to bootstrap a proteomic search. These searches return sequences of proteins with an N-terminal histidine capable of coordinating a metal ion. We exemplify the approach by the discovery and characterisation of four classes of bacterial metalloproteins, including two that we denote as *anglerases* reflecting their potential to capture transition metal ions from the bacterial environment.

Despite advances in protein 3D-structure prediction[1–3], it remains difficult to know a priori from primary amino acid sequences how amino acid side chains come together into the precise 3-dimensional motifs capable of coordinating metal ions. "Lytic" Polysaccharide MonoOxygenases (LPMOs or PMOs) are copper-dependent enzymes which present an exception to this difficulty[4–7]. LPMOs have an immunoglobulin-like fold structure, the edge of which has a flat face[8] which interacts with a polysaccharide (Fig. 1a). Aside from their canonical fold, what distinguishes LPMOs in the context of searching for metalloproteins is their active site. This site contains a single $Cu^{n+}$ ion ($n$ = 1,2) bound in a signature *histidine brace* motif (Fig. 1b)[9], where the principal coordination to the metal is provided by the nitrogen atoms of the amino ($NH_2$) terminus and the imidazole sidechain of a histidine at position 1 of the mature protein, which acts as a chelate for metal ions *regardless* of the higher order structure of the protein.

Variants of the histidine brace also appear in other metalloproteins[10]. Notable examples include copper chaperones, such as CopC[11,12], and the mononuclear $Cu_B$ site in particulate methane monooxygenases[9,13], where, in the latter case, a further coordinating histidine completes the copper coordination sphere (replacing the exogenous ligand, *L*, Fig. 1b)[14]. In some cases, residues such as aspartate, augment the coordination environment[10,11,15]. The recurring presence of the histidine brace across proteins suggests a broader range of as-yet-undiscovered metalloproteins based on the metal-ion chelating properties of N-terminal amino acids: a premise that underpins work described herein.

Functionally, mature LPMOs are secreted proteins, the signal peptides (SPs) of which are cleaved during export. This cleavage exposes the N-terminal amino group, enabling it to coordinate metal ions via lone-pair donation (Fig. 1c). We hypothesised that a SP (selected manually, or using a canonical sequence) followed immediately by a histidine residue could be used as a "bootstrap" sequence to search not only for LPMOs, but also other secreted metalloproteins that contain a histidine residue at their N-terminus. We reasoned that by dint of the N-terminal histidine acting as a natural chelate utilising its $NH_2$ terminus to coordinate to a range of $3d$-transition metal ions,

[1]Department of Chemistry, University of York, Heslington, York, UK. [2]Department of Biology, University of York, Heslington, York, UK. ✉e-mail: sean.sweeney@york.ac.uk; gideon.davies@york.ac.uk; paul.walton@york.ac.uk

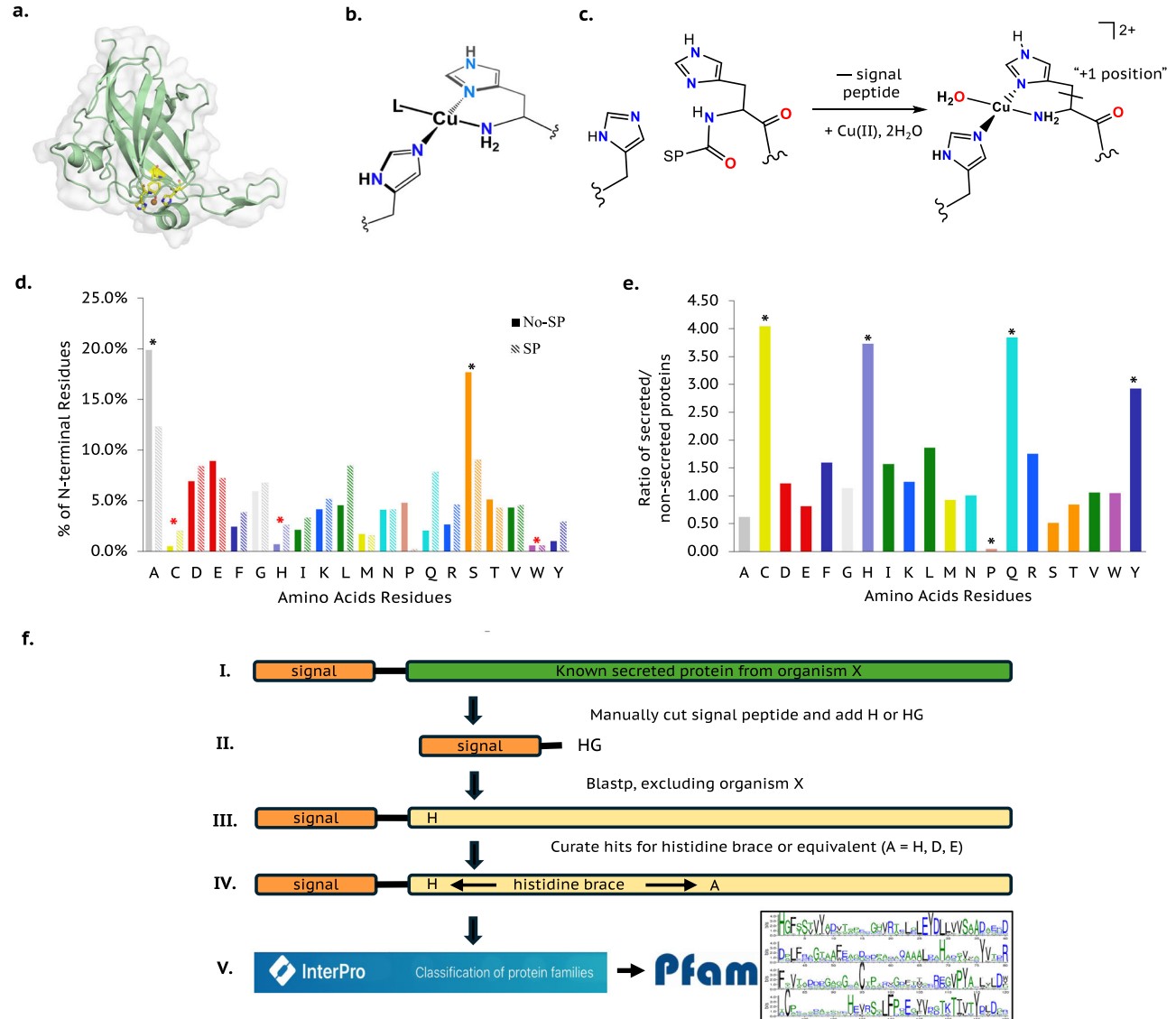

**Fig. 1 | Signal strapping method rationale. a** cartoon view of an AA15 LPMO structure (id:5MSZ)[8] depicting copper active site. **b** copper histidine brace active site (L = exogenous ligand, such as water). **c** cleavage of (SP) from a protein sequence to reveal a free NH₂ group and the N-terminal histidine chelate. **d** frequency of N-terminal amino acids in Eukarya as a fraction of all sequences for intracellular (filled columns) and secreted (diagonal stripes columns). Black asterisks represent the amino acid commonly found adjacent to the N-terminus. Red asterisks represent the amino acid least likely to be found adjacent to the N-terminus. **e** ratio of secreted *vs.* non-secreted N-terminal amino acids as a fraction of all proteins containing that amino acid at the N-terminus. Black asterisks represent the amino acid three to four times more likely to be found adjacent to the N-terminus of a protein with a SP. The amino acid residues were coloured according to the Rasmol "amino" colour scheme. **f** The signal strapping method steps. InterPro and Pfam logos are obtained from the InterPro website and are freely available under the CC0 public domain license. Source data are provided as a Source Data file.

that this approach could provide a step-change in the search for metalloproteins that are targeted to specific cellular or extracellular locations by their SPs.

Here, we demonstrate the utility of the signal strapping method by identifying four metalloprotein families, each of which displays diverse 3-dimensional folds, distinct from that the classical LPMOs. For clarity, we refer to this method as "signal strapping"—a term derived from "bootstrapping," reflecting the concept of initiating discovery using a minimal, adapted sequence.

## Results and discussion

First, we assessed whether signal-peptide-containing (SP) proteins were more likely to expose a metal-coordinating amino acid at the N-terminus after SP cleavage. In doing so, we hypothesised that cytosolic proteins with a free N-terminal histidine would chelate redox-

active metals (e.g., Cu), potentially generating harmful reactive oxygen species via redox-cycling. As such, the expectation is that such proteins would be less prevalent than those that are secreted. To test this hypothesis, we analysed the SignalP 6.0[16] training set comparing amino acid frequencies at position two (post-methionine cleavage) versus position one of the mature protein (post-SP cleavage; Fig. 1d). This analysis shows that alanine is the most common residue adjacent to the N-terminus in both cytosolic (20%) and signal peptide-cleaved proteins (12%), followed by serine (18% and 9%). In contrast, cysteine (0.5%, 2.0%), histidine (0.7%, 2.6%), and tryptophan (0.6%, 0.6%) are the least common (cysteine and histidine are metal-chelating residues). Extending this analysis, Fig. 1e shows that cysteine, histidine, glutamine, proline, and tyrosine occur 3–4 times more frequently at the N-terminus of secreted proteins compared to cytosolic ones. Notably, the first three of these residues are known metal chelators,

**Table 1 | Protein sequences identified through the signal strapping technique, with the sequence used in a Blastp search**

| Protein family identifier | Original SP sequence source | Original protein domain source | Original signal peptide sequence used to search |
|---|---|---|---|
| DUF4198 | OAS21937.1 | GH5 | MRHWIFASLLVLSTPPATA**HG** |
| DUF6702 | WP_086125165.1 | DUF4198 | MKKWIFSSLLVLVSTAQA**HE** |
| *Ang*-1 | WP_230780104.1 | SOD_Ni | MTRLLTATLALMFTASIASA**H** |
| *Ang*-2 | A0A399HGR6 | GH16 | MSGAPRIRRRHPAHRARPRNLRIAVAVATVTGLAAVTLTATAQA**H** |

Manually appended residue (His/H) shown in bold.

supporting our premise that signal peptide cleavage can reveal residues suitable for metal coordination.

Accordingly, we define "*signal-strapping*" (Fig. 1f) as follows: **I** - Select the sequence of a known secreted protein from an organism (e.g., cellulase) and identify the SP using SignalP 6.0 or use a consensus sequence likewise. **II** - Append a histidine (H) or dipeptide (HX, where X is any residue) to the C-terminus of the SP. **III** - Use the modified sequence in a blastp search of the NCBI or Uniprot databases, excluding the original organism. (Blastp at NCBI or Uniprot automatically adjusts search parameters for a 'short sequence'. No other greater adjustments were made to these parameters. Please, see the material and methods section for more details). **IV** - Review hits for conservation of N-terminal histidines and for the presence, although not necessarily conservation, of other potential metal-coordinating residues, e.g., histidine, cysteine, aspartate (an N-terminal histidine is capable of stably chelating a transition metal ion alone without further coordination from other amino acid residues). **V** - Confirm conservation of the histidine following predicted SP cleavage and classify the resulting sequence using InterPro and Pfam.

Using the signal strapping approach, we identified several candidate metalloproteins. Here, we focus on four representative examples (Table 1). Each protein was heterologously expressed, purified, and characterised for metal-binding properties using thermal shift assays (TSA) and spectroscopy. Crystal structures were also determined for selected representatives of the DUF4198 and Anglerase-1 families.

## DUF4198−a secreted nickel-binding protein

We initially selected a signal peptide from a GH5 family protein (GenBank: OAS21937.1) from *Pseudomonas putida* and manually appended "HG" at the C-terminus (Table 1). GH5 proteins are widely secreted across both eukaryotes and prokaryotes, offering broad taxonomic coverage in downstream searches. This sequence was used in a standard blastp search against all organisms excluding *P. putida*. The search recovered mostly glycoside hydrolases from several bacteria, but also sequences of uncharacterised proteins. Amongst these was a DUF4198 domain-containing protein from *Hydrogenophaga sp. IBVHS1* (WP_086125165.1) with an N-terminal histidine. This protein sequence was then used in a search which established that DUF4198 is found broadly distributed in bacteria (mostly *Pseudomonadota*) with some archaeal sequences (Supplementary Fig. 1a) and that these contain, following SP cleavage, a mostly conserved (79%) N-terminal histidine along with other potential metal-coordinating residues further in the sequences, exhibiting good levels of conservation (Supplementary Fig. 1b and Supplementary Data 1).

To infer the potential function, we examined the genomic neighbourhoods (GN) of DUF4198 using the EFI webserver[17]. Adjacent genes included TonB receptors and NikR_C family transcriptional regulators, both linked to nickel metabolism (Supplementary Fig. 2a). This association is further supported by transcriptomic data showing DUF4198 expression within nickel and cobalt ATP-binding cassette transporter operons[18]. Based on this evidence, we selected a DUF4198 protein from *Ideonella sakaiensis* (WP_082368692.1, "*Is*DUF4198") for recombinant expression. Though *I. sakaiensis* is known for plastic degradation[19], genes involved in that process (e.g., PETases, MHE-Tases) are not linked to DUF4198 (Supplementary Fig. 2b).

Purified *Is*DUF4198 (Supplementary Fig. 3) was assessed for its metal-binding capacity using TSA. At a 1:1 protein:metal molar ratio, the greatest thermal stabilisation was observed with $Ni^{2+}$ ($\Delta T_m = +7.0\,^\circ C$, Fig. 2a), followed by $Zn^{2+}$ ($+4.6\,^\circ C$), $Co^{2+}$ ($+4.0\,^\circ C$) and $Cu^{2+}$ ($-0.2\,^\circ C$). These relative shifts, save for $Cu^{2+}$, are in accord with those expected from the Irving-Williams series. Further TSA analysis at multiple stoichiometries (200:1 and 10:1, Supplementary Fig. 4) and a measured dissociation constant for $Ni^{2+}$ ($K_d = 18 \pm 3\,nM$; Supplementary Fig. 5) confirm nickel binding, consistent with the GN analysis described above. The potential role for DUF4198 in metal transport rather than as an active metalloenzyme finds further support from assays described herein (Methods), which showed only weak peroxidase activity of isolated Ni-*Is*DUF4198 (Supplementary Discussion and Supplementary Fig. 6).

The *Is*DUF4198 protein was co-crystallised with $Ni^{2+}$ and its structure resolved at 1.4 Å resolution using molecular replacement (Supplementary Table 1). The structure reveals a β-sandwich fold composed of thirteen β-strands connected by flexible loops (Fig. 2b) (PDB: 9GCB). Structural comparison using the DALI server identified a cytoplasmic sulphur-carrier protein from *Chlorobium limicola* (PDB ID: 2NNC)[20] as the closest match, though it shares only 15% sequence identity and lacks an N-terminal histidine (Z-score: 5.1).

The *Is*DUF4198 structure features a cleft measuring ~12 × 20 Å, which houses the metal-binding site (Fig. 2c, d). As anticipated, a $Ni^{2+}$ ion is located within this positively charged cleft (Fig. 2e), coordinated by the N-terminal histidine and the τN-atom of a conserved His18 residue−both of which are conserved across homologues (Fig. 2f). Nickel coordination is completed by a water molecule and a chelating malonate (from crystallisation buffer), resulting in near-octahedral coordination geometry (Supplementary Table 2). Unlike the canonical histidine brace found in LPMOs, the three nitrogen atoms of the coordinating histidine groups are arranged such that they facially cap one half of the metal ion coordination sphere (Fig. 2c, d). Unlike the canonical T-shaped histidine brace seen in LPMOs, the three nitrogen donors in *Is*DUF4198 form a facially capping arrangement around the $Ni^{2+}$ ion (Fig. 2c, d). This coordination geometry likely underlies the protein's preference for $Ni^{2+}$ over $Cu^{2+}$−a notable inversion of the Irving−Williams series[21].

## DUF6702−a metal sequesterase

Building on the discovery of Ni-containing DUF4198 proteins and their association with metal-dependent biochemistry, we selected the SP of the DUF4198-domain protein from *Hydrogenophaga* sp. IBVHS1 for signal strapping (Table 1). A blastp search (excluding *Hydrogenophaga* sp.) retrieved DUF4198 homologues and several uncharacterised proteins. From the latter, a DUF6702 family protein from *Pseudoalteromonas* sp. A757 (WP_128727020) was identified, containing an N-terminal histidine residue. Analysis of DUF6702 family members in the InterPro/Pfam databases revealed that 65% of sequences retain an N-terminal histidine after predicted SP cleavage, and potential metal-coordinating amino acids are present in the rest of the sequences, exhibiting variable levels of conservation (Supplementary Fig. 7a and Supplementary Data 2). This domain appears to be exclusive to bacteria, particularly within the Bacteroidota phylum (Supplementary Fig. 7b).

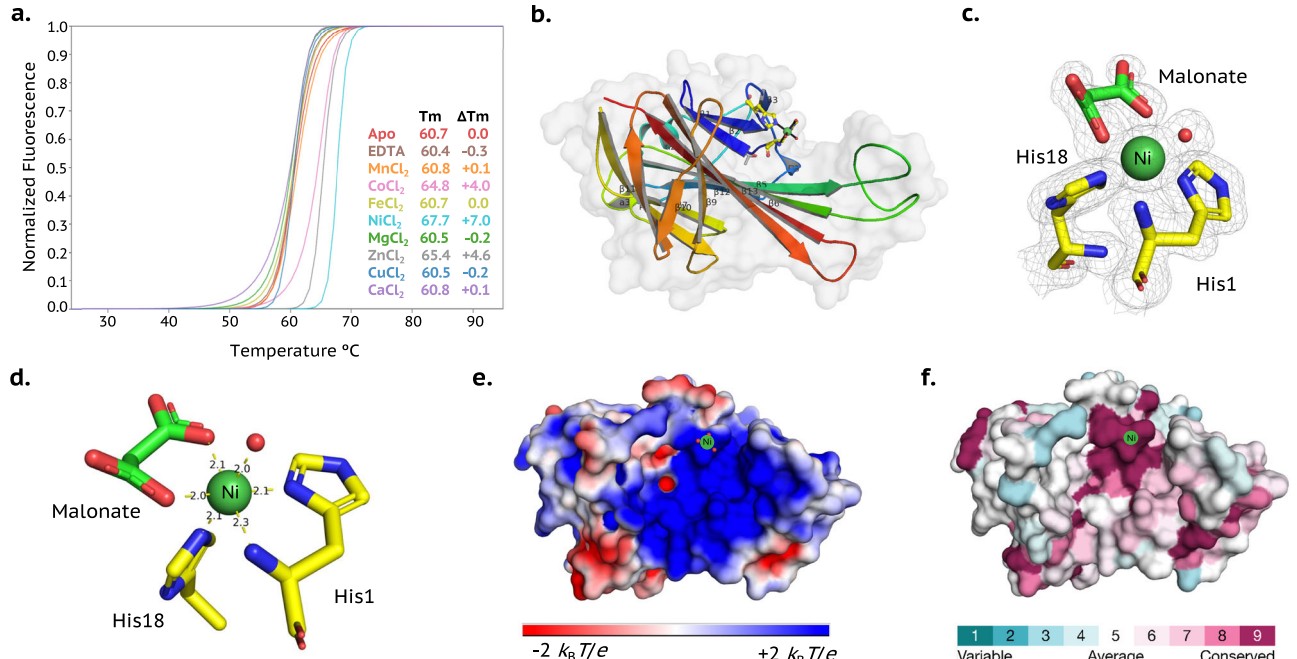

**Fig. 2 | Structure and depictions of Ni-containing *Is*DUF4198. a** TSA of 15 μM *apo* *Is*DUF4198 in the presence of several metals at 15 μM (1:1 [metal:protein] molar equivalent). Each TSA curve is coloured according with its respective metal-Cl₂ as shown in the figure legend. Positive shifts indicate ligand binding to the proteins and thermal stabilisation, while negative shifts indicate ligand binding to the proteins and thermal destabilisation due to protein aggregation or precipitation. **b** overall protein structure showing secondary structural elements, Van-der-Waals surface (light grey) and amino acid residues/water molecules coordinating to the nickel ion (coloured cylinders). **c** Electron density at 1.5 RMSD in the non-canonical histidine brace from *Is*DUF4198. $Ni^{2+}$ centre is in green, waters in red. **d** Distances among the $Ni^{2+}$ centre to neighbour nitrogens from the histidines. **e** Electrostatic surface potential of *Is*DUF4198 generated by the APBS plugin for PyMol settled at pH 7.0 to calculate and visualise the surface electrostatic potential at $\pm 2 \, k_BT/e$, showing the positively charged cleft where the $Ni^{2+}$ binding site is located. **f** Sequence conservation analysis (ConSurf) of *Is*DUF4198, viewing towards the cleft at the protein surface, displaying the high conservation of the histidine brace. The coloured surface is based on the ConSurf score. Source data are provided as a Source Data file.

To investigate the potential biological function of DUF6702, we analysed its GN. While many neighbouring genes encode proteins of unknown function (Supplementary Fig. 8a), several encode M1 peptidases and HAD_2 domain-containing proteins, the latter is predicted to be haloacid dehalogenase-like hydrolases. M1 peptidases utilise different metal ions—zinc, cobalt, manganese or copper—within their active sites[22]. Similarly, HAD superfamily enzymes (for example, phosphatases, phosphonatases, P-type ATPases, and beta phosphoglucomutases) also rely on metal ions for their function[23]. This context supports the hypothesis that DUF6702 is metal-associated.

Given its likely metalloprotein nature, we selected a DUF6702 protein from pathogenic bacterium *Pseudomonas aeruginosa* (VZT40374; hereafter *Pa*DUF6702) for recombinant expression in *E. coli* (Supplementary Fig. 9). The neighbouring genes are depicted in Supplementary Fig. 8b. TSA analysis at a 1:1 molar ratio revealed a large stabilising shift upon $Ni^{2+}$ binding ($\Delta T_m = +17.2 \, °C$), with notable shifts also observed for $Co^{2+}$ ($+10.7 \, °C$) and $Cu^{2+}$ ($+7.0 \, °C$) (Fig. 3a and Supplementary Fig. 10). Dissociation constants (Supplementary Fig. 11) confirmed high affinities: $Ni^{2+}$ ($5.60 \pm 0.1$ pM), $Co^{2+}$ ($3.6 \pm 0.2$ nM), and $Cu^{2+}$ ($3.26 \pm 0.11$ nM). These findings support the designation of *Pa*DUF6702 as a metal sequesterase with a particular affinity for $Ni^{2+}$. Importantly, oxidase and peroxidase assays showed no significant activity for Ni-, Co-, or Cu-bound forms (Supplementary Fig. 6 and Supplementary Discussion), suggesting a non-enzymatic metal-binding or transport function.

Crystallisation of metal-bound *Pa*DUF6702 was unsuccessful. Therefore, structure prediction was performed using AlphaFold3 (AF3). Unfortunately, AF3 is not parameterised for Ni-containing proteins, so as alternatives, five models (0–4) were generated for the *apo*, Cu-loaded, and Co-loaded states. All models exhibited high global

pLDDT confidence scores (Supplementary Table 3). One of which, model 3, was selected based on metal ion structure parameters described in the supplementary results (Supplementary Fig. 12 and Supplementary Tables 4 and 5). The overall structure of the *Pa*DUF6702 model displayed a beta-sandwich architecture with an immunoglobulin-like fold, composed of nine beta-strands connected by several loops (Fig. 3b). In the *apo* form, the second beta-strand connects with the third through two alpha-helices, whereas the Cu-bound form exhibits three alpha-helices, suggesting a structural change upon metal binding (Supplementary Fig. 12a–c).

The *Pa*DUF6702 structure features a partially flat surface at the predicted metal-binding site, resembling that of some LPMOs[9] and metal-binding carrier proteins[24] (Fig. 3b). In both Cu- and Co-bound models, the predicted site comprises three histidine residues: His1, His28, and His32 (Supplementary Fig. 13a, b and Supplementary Discussion). The coordination involves the N-terminal His1, consistent with the expectations of the signal strapping approach. However, in the *apo* form, the side chain of His1 is modelled with low confidence (Supplementary Fig. 13c and Supplementary Discussion).

As no crystal structure was available, EPR spectroscopy was used to probe the metal coordination geometry of Cu-*Pa*DUF6702 at pH 7.0. X-band CW-EPR of $Cu^{2+}$-*Pa*DUF6702 (Fig. 3d) gave spin-Hamiltonian parameters with $g_3 > g_2 \sim g_1$ ($g_1$ 2.050, $g_2$, 2.065, $g_3$ 2.256 and $A_3$ 550 MHz), consistent with a near square planar coordination geometry, supporting the selected structural model (Fig. 3c, d). Electrostatic surface potential (ESP) calculations of *Pa*DUF6702 at pH 7.0 revealed a positively charged metal-binding site, indicating a potential docking site for a protein partner or substrate (Fig. 3e). ConSurf analysis further confirmed the conservation of His1 and adjacent residues across homologous sequences (Fig. 3f). These structural features are consistent with the DUF6702 family acting as a metal transport

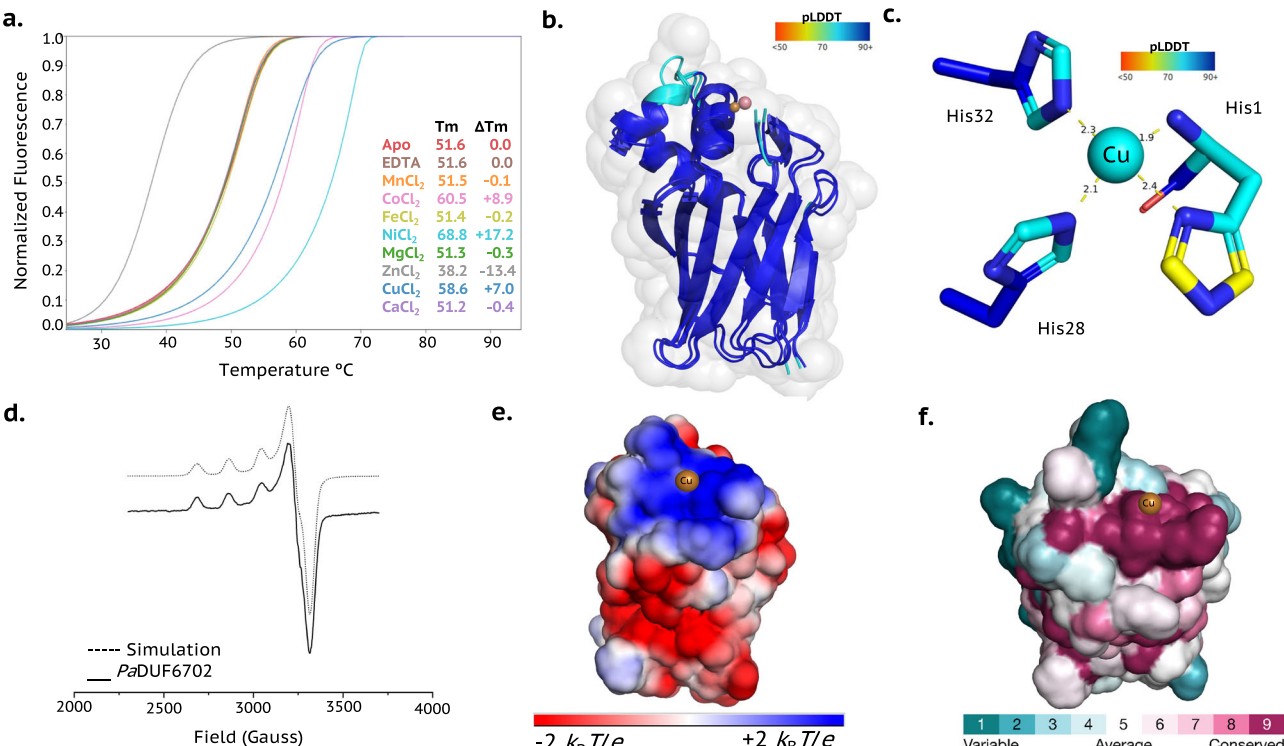

**Fig. 3 | Structure and depictions of Cu-containing DUF6702. a** TSA of *apo PaDUF6702a* in the presence of several metals at 1:1 molar equivalent. Each TSA curve is coloured according with its respective metal-Cl$_2$ as shown in the figure legend. Positive shifts indicate ligand binding to the proteins and thermal stabilisation, while negative shifts indicate ligand binding to the proteins and thermal destabilisation due to protein aggregation or precipitation. **b** Overlapped *Apo-*, Co- and Cu-forms modelled structures of *PaDUF6702a*, showing secondary structural elements coloured by the pLDDT scores; Van der Waals' surface (light grey). Cu$^{2+}$ centre is coloured in brown and cobalt centre coloured in light pink **c**. Metal binding site of non-canonical tri-histidine brace from *PaDUF6702*. Cu$^{2+}$ centre is coloured by the pLDDT score. **d** EPR spectra from *PaDUF6702* at pH 7.0. **e** Electrostatic surface potential of *PaDUF6702a* generated by the APBS plugin for PyMol settled at pH 7.0 to calculate and visualise the surface electrostatic potential at ± 2 $K_B T/e$. The Cu$^{2+}$ centre is coloured in brown. **f** Sequence conservation analysis (ConSurf) of *PaDUF6702a* coloured by the ConSurf score. The Cu$^{2+}$ centre is coloured brown. Source data are provided as a Source Data file.

protein, insofar as the metal binding site can potentially directly interact with other proteins.

## Ang-1 and Ang-2 − metal "anglerases" appended to membrane-bound permeases

We applied the signal strapping method using the SP from a nickel superoxide dismutase (Pfam 09055, WP_230780104.1) of *Roseiconus lacunae* (Table 1). Nickel superoxide dismutases are known to coordinate Ni$^{2+}$ via an N-terminal histidine. This search retrieved many Ni-superoxide dismutases alongside proteins of unknown function and cytochrome c and copper chaperones (CopC) proteins; the last of which is known for an N-terminal histidine that binds copper[25]. In addition, further sequences annotated as putative hydrogenase/urease accessory proteins (HupE/UreJ-2) were found with an N-terminal histidine after the SP. These last domains are known nickel permeases[26].

We selected a representative HupE/UreJ-2 protein (Pfam 13795, InterPro IPR032809) from a *Rhodobacteraceae bacterium* (ETA49561.1) and analysed it using InterPro to evaluate the protein domain conservation. The search revealed that the query protein contains two separate domains: a non-cytoplasmic domain containing the N-terminal histidine (hereafter termed Ang-1), and a C-terminal transmembrane (TM) HupE/UreJ-2 domain. A BLAST search using this full sequence confirmed the two-domain architecture (Supplementary Fig. 14a). Further analysis with ConservFold showed 100% conservation of the N-terminal histidine following SP cleavage, along with highly conserved potential metal-coordinating amino acids in other parts of the sequences (Supplementary Fig. 14b and Supplementary Data 3).

In a separate search, we used the SP of a GH16 family hemicellulase (IPR000757; SUPfam 49899; WP_230780104.1) from

*Streptomyces* sp. YIM 130001 (Table 1). In contrast to the known metalloprotein SP used for Ang-1, GH16s are broadly secreted across bacteria and eukaryotes. A BLASTp search (excluding *Streptomyces* sp.) returned GH16 proteins, metallopeptidase inhibitors (I36), cellulases, and again, proteins with a C-terminal HupE/UreJ-2 domain, with N-terminal domains distinct from Ang-1 (hereafter Ang-2). Like Ang-1, Ang-2 displays a non-cytoplasmic domain after the SP, followed by the TM HupE/UreJ-2 at the C-terminal (Supplementary Fig. 15a), and ConservFold results showed 100% conservation of His1 immediately following SP cleavage, along with highly conserved potential metal-coordinating amino acids in other parts of the sequences (Supplementary Fig. 15b and Supplementary Data 4).

As both Ang-1 and Ang-2 possess similar C-terminal HupE/UreJ-2 domains (42% of identity), we also performed full-length and only Ang domain sequence alignments (in Uniprot-CLUSTAL) to assess their similarity. The two full-length proteins share 34% of sequence identity (Supplementary Fig. 15c), while only the Ang domains share 24% identity, suggesting distantly related for the different Ang domains. Ang-1 proteins are found exclusively in bacteria, particularly within the Pseudomonadota phylum, while Ang-2 sequences are restricted to Actinomycetota (Supplementary Fig. 15d).

To gain insight into the biological functions of Ang-1 and Ang-2, we analysed their genomic neighbourhoods. For Ang-1, the most frequently co-occurring genes encode proteins of unknown function (Fig. 4a). DUF4198 domains (see above), Indigoidine A-like proteins (InterPro IPR022830), and PfkB kinases (Pfam 00294) were also prevalent. Indigoidine synthase proteins are known to exhibit glycosidase activity, requiring manganese in the active site[27,28]. For Ang-2, proteins with unknown functions were again the most frequently encountered

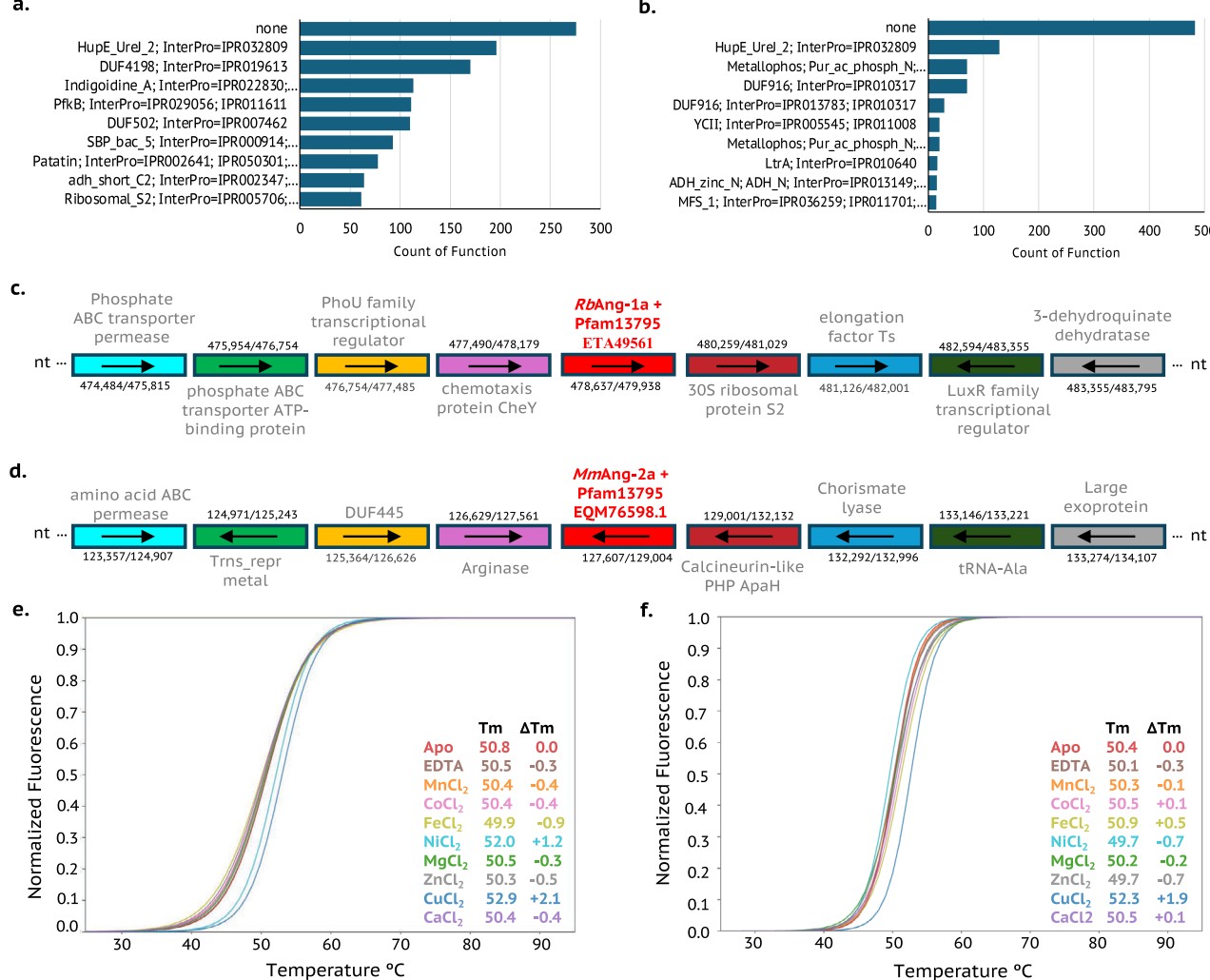

**Fig. 4 | Domain neighbourhoods, genomic organisation and metal binding analysis of Ang-1 and Ang-2 domains. a** Counts of function for the *Ang*-1 + Pfam13795 neighbour genes across bacterial genomes from the RefSeq database. **b** Counts of function for the Ang-2 + Pfam13795 neighbour genes across bacterial genomes from the RefSeq database. **c** Genomic organisation of Ang-1-containing operon in the genome of *R. bacterium*. **d** Genomic organisation of Ang-2-containing operon in the genome of *M. maritypicum*. **e** TSA of *apo RbAng-1a* in the presence of several metals at 1:1 molar equivalent. **f** TSA of *apo MmAng-2a* in the presence of several metals at 1:1 molar equivalent. Each TSA curve is coloured according with its respective metal-$Cl_2$ as shown in the figure legend. Positive shifts indicate ligand binding to the proteins and thermal stabilisation, while negative shifts indicate ligand binding to the proteins and thermal destabilisation due to protein aggregation or precipitation. Source data are provided as a Source Data file.

(Fig. 4b), followed by metallophosphatases and DUF916 domains. Metallophosphatases possess a conserved bimetallic active site (typically Mn, Fe, or Zn)[29]. DUF916 proteins (Pfam 06030) have been recently implicated as WxL-interacting proteins (WxLIPs), serving as scaffolds for anchoring WxL domains to bacterial cell wall peptidoglycan[30]. For both Ang-1 and Ang-2, therefore, the gene neighbourhoods encode some metal-dependent proteins.

The Ang-1 domain of the HupE/UreJ-2 protein from *R. bacterium* (hereafter *RbAng-1a*), excluding its transmembrane (TM) region, was cloned for recombinant expression in *E. coli* (Supplementary Fig. 16). The corresponding gene locus and neighbouring genes are shown in Fig. 4c. Similarly, the Ang-2 domain from *Microbacterium maritypicum* MF109 (*MmAng-2a*), also lacking the TM region, was expressed in *E. coli* (Supplementary Fig. 17), with its genomic context shown in Fig. 4d.

TSAs were used to assess metal binding by Ang-1 and Ang-2. At a 1:1 molar ratio, the largest increase in melting temperature ($\Delta Tm = +2.1\,°C$) was observed with $Cu^{2+}$, followed by $Ni^{2+}$ ($+1.2\,°C$). Other metals caused modest destabilisation: $Fe^{2+}$ ($-0.9\,°C$), $Zn^{2+}$ ($-0.5\,°C$), $Mn^{2+}$ ($-0.4\,°C$), and $Co^{2+}$ ($-0.4\,°C$) (Fig. 4e and Supplementary Fig. 18). Based on these results, *RbAng-1a* is designated a Cu-

metalloprotein, though $Ni^{2+}$ binding is also significant. Dissociation constants (Kd) determined by TSA were $73.6 \pm 2.3\,nM$ for $Cu^{2+}$ and $220 \pm 10\,nM$ for $Ni^{2+}$ (Supplementary Fig. 19), suggesting physiological relevance for both ions.

*MmAng-2a* was also assessed for metal-binding capacity (Fig. 2f). The largest thermal shift at 1:1 metal:protein ratio occurred with $Cu^{2+}$ ($+1.9\,°C$), followed by $Fe^{2+}$ ($+0.5\,°C$) (Supplementary Fig. 20). The dissociation constant for $Cu^{2+}$ was $62.3 \pm 2.8\,nM$ (Supplementary Fig. 21), supporting its classification as a Cu-binding metalloprotein. $Fe^{2+}$ binding may also be relevant, consistent with the metal dependencies of nearby genes in its operon.

Oxidase and peroxidase assays of metal-loaded *RbAng-1a* and *MmAng-2a* (M = Cu, Co, or Ni) revealed no substantial activity relative to LPMOs, though the Cu-loaded forms showed weak oxidase activity (Supplementary Fig. 6 and Supplementary Discussion). These levels are comparable to those of free $Cu^{2+}$ and known weak LPMO oxidases, suggesting that the primary function of Ang-1 and Ang-2 is metal binding or transport, though a modest enzymatic role cannot be fully excluded.

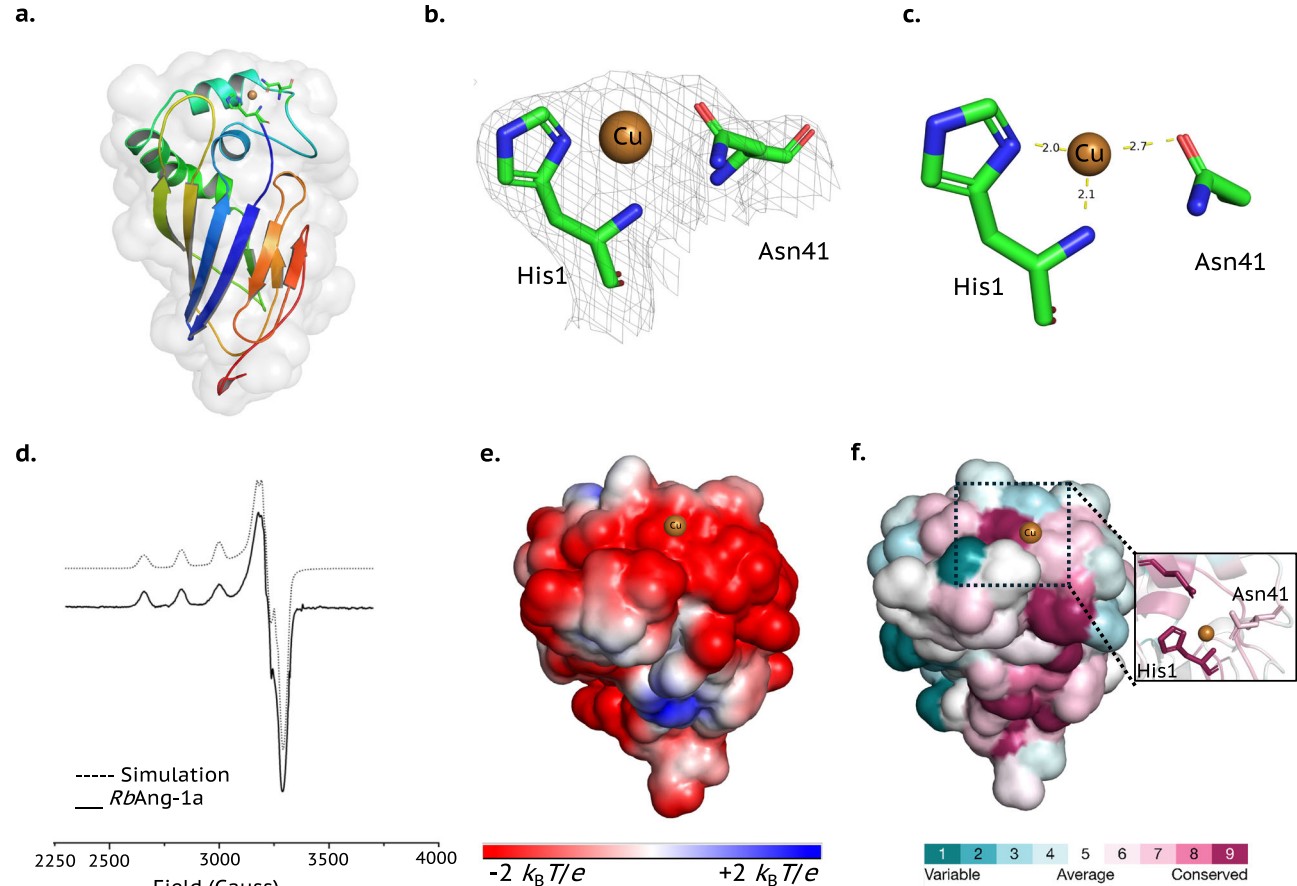

**Fig. 5 | Structures of Cu-containing RbAng-1a. a** overall protein structure showing secondary structural elements, Van der Waals' surface (light grey) and amino acid residues molecules coordinating to the copper ion (green coloured cylinders); **b** Electron density at 1.0 RMSD for the metal-binding site of *Rb*Ang-1a. Cu⁺ centre is in brown. **c** Distances of the Cu centre to nitrogens in His1 and oxygen from Asn41. **d** X-band, CW-EPR spectrum (150 K) of Cu²⁺-*Rb*Ang-1a. **e** Electronic environment surrounding the metal-binding site of *Rb*Ang-1a generated by the APBS[53] plugin for PyMol[50] settled at pH 7.0 to calculate and visualise the surface electrostatic

potential at ± 2 $K_B$T/e. This analysis showed that the top flat surface harbouring the active site is negatively charged. Cu⁺ centre is in brown. **f** The conservation of the His1 is displayed by the Consurf[55] analysis, further emphasising the importance of the mono-histidine brace within this herein identified metalloprotein family. However, the surrounding amino acids are not highly conserved, even the coordinating asparagine 41. Cu⁺ centre is in brown. Source data are provided as a Source Data file.

*Rb*Ang-1a was crystallised in the presence of Cu²⁺ and its structure resolved to 1.9 Å using molecular replacement with an AlphaFold2 (AF2[31]) model (Supplementary Table 6). The protein adopts a beta-sandwich architecture with an immunoglobulin-like fold, composed of eight ß-strands connected by several loops (Fig. 5a) (PDB id:9GCE), somewhat resembling *Pa*DUF6702 (RMSD of ~ 6.7 Å, Supplementary Fig. 22). The second beta-strand connects to the third via three alpha-helices. A DALI search identified the closest structural match as a zinc-binding protein from *Vibrio cholerae* (PDB ID: 8F1B)[24].

The copper-binding site of *Rb*Ang-1a lies on a flat surface (Fig. 5a, b), where Cu⁺ is coordinated by the N-terminal His1 and the O atom of Asn41 in a T-shaped geometry (Fig. 5c). This coordination motif has been reported as a 'mono-histidine brace'[32,33]. The *Rb*Ang-1a Cu-ligand bond lengths range from 2.1 to 2.7 Å (Supplementary Table 7), and the copper ion is three-coordinate. The geometry suggests a Cu⁺ oxidation state, following photoreduction of Cu²⁺ in the X-ray beam. To verify that the site can also support Cu²⁺, we performed X-band CW-EPR spectroscopy at pH 5. The resulting spin-Hamiltonian parameters of $g_3 = 2.28$ and $A_3 = 525$ MHz and superhyperfine couplings from two Cu-N interactions $A_N = 35, 42$ MHz (Fig. 3d) are consistent with Cu²⁺-*Rb*Ang-1a binding Cu²⁺ in a T-shaped coordination from the protein-based ligands (N₂O coordinating atoms), augmented by a coordinating ligand, likely an exogenous water molecule. This would

yield an overall four-coordinate (N₂O₂) near-square planar geometry at the Cu²⁺ (Fig. 5c).

We also modelled the full-length *Cu*-RbAng-1a-HupE/UreJ-2 complex using AF3[3] which achieved a high confidence prediction pLDDT score (> 90.0) (Supplementary Fig. 23a). The model closely aligns with the crystal structure (RMSD ~ 0.3 Å; Supplementary Fig. 23b), though discrepancies were noted in the coordination geometry at the metal ion. Whereas the AF3 model shows bond lengths of 1.9 to 3.4 Å with a pyramidal geometry at the Cu, while the crystal structure has 2.0 to 2.7 Å and a T-shaped geometry (RMS difference 0.14 Å, Supplementary Fig. 24). Furthermore, the AF3 prediction did not match the observed EPR-derived parameters. These findings underscore the value of AF3 for overall fold prediction while highlighting its limitations in accurately modelling metal coordination.

Crystallisation of Cu-loaded *Mm*Ang-2a was unsuccessful despite extensive screening. Therefore, structural modelling was performed using AF3 for both apo and Cu-bound states. Five models were generated for each condition, with high pLDDT scores (≤ 80). Model 4 (apo) and Model 1 (Cu-bound) were selected as the most reliable (Supplementary Table 8). The predicted *Mm*Ang-2a-HupE/UreJ-2 structure comprises a β-sandwich *Mm*Ang-2 domain and a HupE/UreJ-2 domain containing eight α-helices (Supplementary Fig. 25a).

Two candidate metal binding sites were identified, one in the *Mm*Ang-2 domain (His1/Asp33) (Supplementary Fig. 25b) and another in the HupE/UreJ-2 domain (His194, His201, His229) (Supplementary Fig. 25c). Notwithstanding the overall high confidence in the structure, however, some uncertainty in the His1 site of the *Mm*Ang-2 domain was evident with a low pLDDT score and lower inter-chain prediction TM-score (Supplementary Table 9). In addition, the *Mm*Ang-2a predicted metal binding exhibited bond lengths outside of normal ranges, suggesting again that AF3 metal coordination prediction must be interpreted cautiously.

Given this uncertainty, to validate Cu binding experimentally, EPR spectroscopy was conducted at pH 5.0. The resulting spin-Hamiltonian parameters for the Cu of $g_3 = 2.27$ and $A_3 = 530$ MHz, along with superhyperfine couplings from two Cu-N interactions $A_N = 45, 35$ MHz (Supplementary Fig. 25d), are consistent with the coordination geometry seen in *Rb*Ang-1a, showing that *Mm*Ang-2a is a bona fide metalloprotein, and also that the detailed modelling of the metal binding site by AF3 is likely inaccurate. ESP analysis of *Mm*Ang-2a at pH 7.0 revealed several charged side chains around the mono-histidine brace (Supplementary Fig. 25e), with negative charges dominating the metal-binding site. Consurf analysis showed that His1 and Asp33 are highly conserved among Ang-2 sequences (Supplementary Fig. 25f).

Based on structural, biochemical, and genomic data presented here, we propose that both Ang-1 and Ang-2 function as metal-binding domains directly linked to membrane-anchored HupE/UreJ-2 permeases. This architecture suggests their involvement in metal capture and assimilation in bacteria.

Using the validated AF3 model of Ang-1 (which aligns closely with its crystal structure), we modelled the full-length Ang-1- and Ang-2-HupE/UreJ-2 apparatus (Fig. 6). In both cases, the metal-binding domain faces the HupE/UreJ-2 domain in a compact "closed" conformation relative to the periplasm (Fig. 6a, b). This is consistent with the modelling context, which lacks solvent and membrane constraints, and thus steers the model to maximise protein-protein interactions, favouring close domain interactions.

A flexible linker region (residues 176–184 in *Rb*Ang-1a, 175–180 in *Mm*Ang-2a) connects the anglerase and HupE/UreJ-2 domains. Flexibility analysis (Supplementary Fig. 26) and normal mode simulations (Supplementary Fig. 27) revealed significant conformational dynamics. These models suggest that the anglerase domain can rotate away from the channel "mouth" of the permease (Supplementary Movie 1), exposing its N-terminal His residue to the surrounding *milieu*, from which it can sequester any adventitious transition metal ions. Indeed, both conformations are modelled by AF3, which further predicts low likelihoods of homo-multimeric complexes for both Ang-1- and Ang-2-HupE/UreJ-2 (Supplementary Fig. 28).

This open–closed transition resembles systems used for glycan acquisition in Bacteroidetes, where a periplasmic receptor feeds substrates into membrane channels[34], but until now, not proposed for metal ion uptake systems. Put more colloquially, the *anglerase* domains 'fish' for metal ions that are fed into the mouth of the permease, thus forming part of a mechanism for transition-metal ion capture and uptake by both in Gram-positive and Gram-negative bacteria. This model is consistent with previous knockout studies showing the essentiality of HupE/UreJ-family transporters in metal homeostasis in nitrogen-fixing bacteria[26]. In this context, we thus suggest that Ang-1 and Ang-2 are named *anglerases*.

## Phylogenetic relationships among the herein identified metalloproteins

To explore the evolutionary context of the herein identified metalloproteins, we performed a phylogenetic analysis (Fig. 6c). Sequences containing DUF4198 form a distinct clade with a high bootstrap value (100), consistent with their divergent fold compared to the other protein families described herein. DUF6702 sequences also form a strong clade (bootstrap 92), but with a shorter phylogenetic distance to the anglerases, reflecting their shared β-sandwich architecture. In addition, a separate DUF6702 subgroup (bootstrap 91) likely represents a subfamily with a divergent fold. The anglerases cluster within four different clades, all with high bootstrap support (75, 99, 97, 92). This result indicates that anglerases, exemplified here by *Rb*Ang-1a and *Mm*Ang-2a, may constitute a broad and structurally diverse family of metalloproteins with variable metal binding preferences and selectivity.

Signal strapping is a protein-sequence search technique that identifies mature, secreted proteins that have amino acids at the N-terminus capable of chelating metal ions. The approach begins with a known signal peptide (consensus sequences could be used), which is manually appended with a metal-coordinating residue (e.g., histidine), and used to bootstrap a proteomic search. Following removal of the signal peptide, the appended residue becomes the first residue of the mature protein and can coordinate transition metals via its amino group and side chain (e.g., imidazole in histidine). As such, amino acid chains with an N-terminal histidine are likely metalloproteins.

This strategy is generalisable to other chelating N-terminal residues, such as cysteine and aspartate. Herein we exemplified this approach for the discovery and characterisation of four unknow metalloproteins. Structural and spectroscopic analyses confirmed that these proteins bind transition metals ($Cu^{2+}$ or $Ni^{2+}$) via N-terminal histidines. Genomic context further revealed that two of these families —designated Ang-1 and Ang-2—are likely components of bacterial metal acquisition systems. These proteins, which we term anglerases, appear to "fish" for metal ions in the extracellular space and deliver them to membrane-bound permeases, highlighting a potentially widespread mechanism for metal ion uptake in bacteria.

## Methods
### Signal strapping methodology
The signal strapping method was performed manually without the support of automatic pipelines or algorithms for the discovery of N-terminal histidine-containing proteins. Therefore, the manual signal strapping pipeline was developed as follows:

**Step 1**: A single or consensus amino acid sequence of a known secreted protein from an organism was selected, and its signal peptide (SP) was identified using SignalP-6.0 (https://services.healthtech.dtu.dk/services/SignalP-6.0/)[16]. The SP prediction was carried out in "slow" model mode, as precise region borders were required, with the output format set to "long output". The organism type was selected based on the taxonomic origin of the SP sequence. The use of a consensus sequence as a search sequence reduces the human bias, which may affect the selection of the sequence and the metalloproteins discovered from such a sequence.

**Step 2**: After identifying the exact SP sequence (or a consensus SP sequence) in the known secreted protein, a histidine (H/His) amino acid was manually appended to the C-terminal end of the SP sequence, followed by any amino acid or none.

**Step 3**: The artificial SP with the appended histidine residue was used as a query in a BLAST search against the NCBI[35] or UniProt (https://www.uniprot.org/blast)[36] databases. For the NCBI search, the Protein BLAST (protein-to-protein) tool was selected, and the SP + H"X" sequence was entered into the query field. The search was run using standard settings, with the non-redundant protein sequence (nr) database. The organism field was set to the original organism of the SP, and the "exclude" option was applied. The algorithm was kept as "blastp," and the maximum number of target sequences was adjusted from 100 to 500 in the algorithm parameters. All other settings remained at their default values.

For the UniProt search, the SP + H"X" sequence was also entered into the query field. The target database was left as default (UniProtKB reference proteomes + Swiss-Prot), without excluding the taxonomic

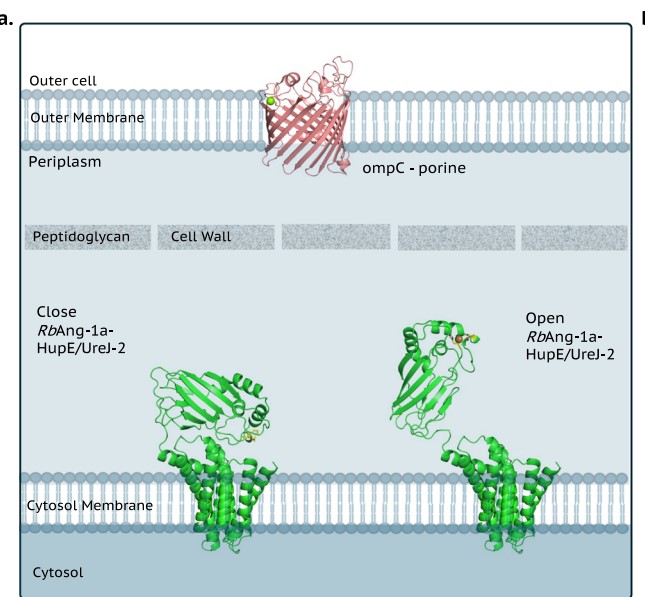

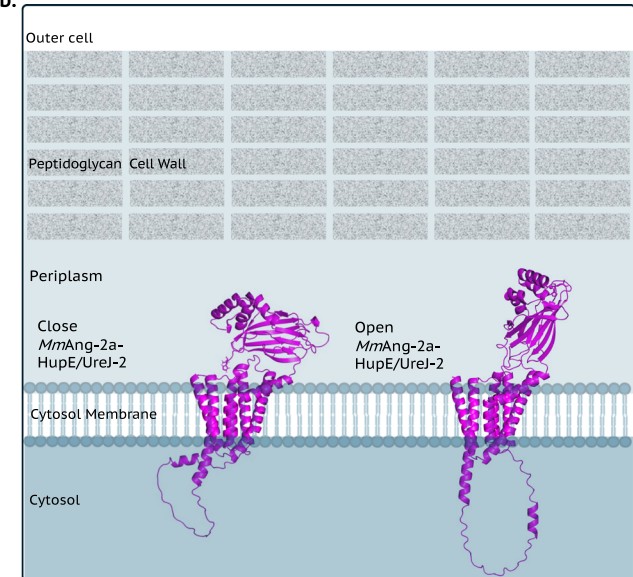

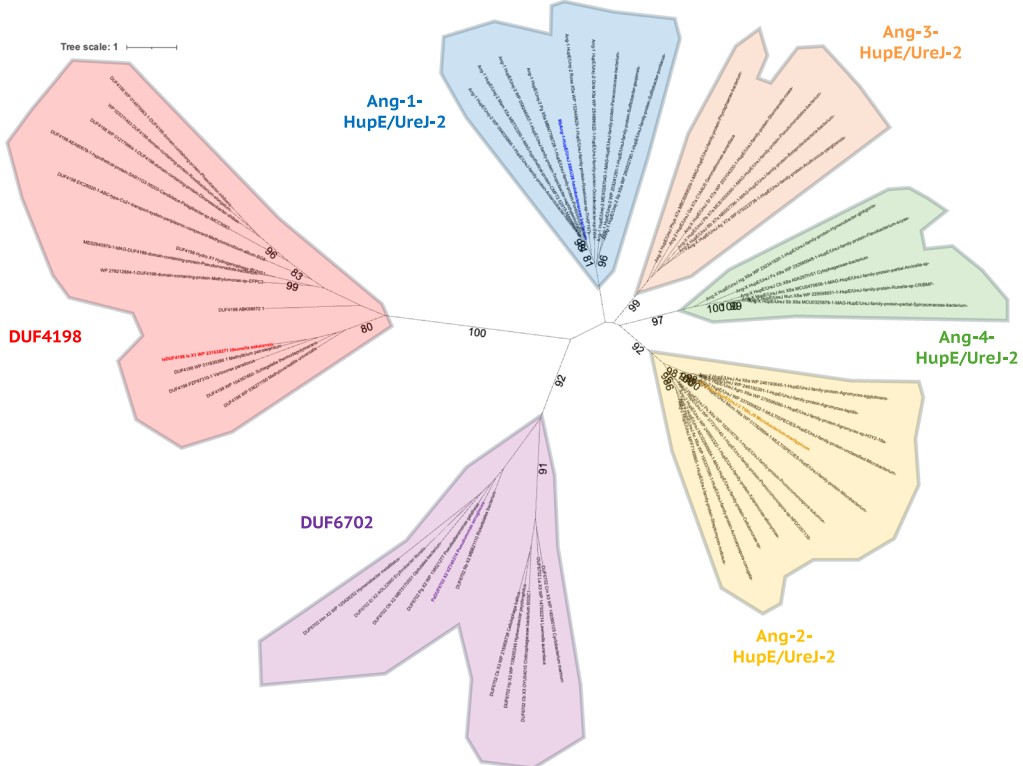

**Fig. 6 | Proposed biochemical function of *anglerases* and its phylogenetic relationships with the DUF4198 and DUF6702. a** Overall full-length modelled structure of *Rb*Ang-1a-HupE/UreJ-2 (green) in the close and open positions on the context of the membrane/cell wall system of gram-negative bacteria. Analysis of *Rb*Ang-1a-HupE/UreJ-2 CDS in the DeepLoc server[67] revealed a putative localisation of the TM HupE/UreJ-2 domain at the cytosol membrane of gram-negative bacteria, and *Rb*Ang-1a at its periplasm. **b** Overall full-length modelled structure of *Mm*Ang-2a-HupE/UreJ-2 (magenta) in the closed and open positions on the context of the membrane/cell wall system of gram-negative bacteria. Analysis of *Mm*Ang-2a-HupE/UreJ-2 CDS in the DeepLoc server[67] revealed a putative localisation of the TM HupE/UreJ-2 domain at the cytosolic membrane of gram-positive bacteria, and *Mm*Ang-2a at the outer cell space. **c** Phylogenetic tree of the herein identified metalloproteins. *Is*DUF4198 is in bold red, followed by *Pa*DUF6702 in purple, *Rb*Ang-1a in blue and *Mm*Ang-2a in yellow. Source data are provided as a Source Data file.

group of the original SP. The advanced parameters remained in default mode, except for the "hits" option, which was adjusted from 250 to 500. Both NCBI and UniProt BLAST tools automatically adjusted the search parameters for short sequences after the search was initiated, and the analysis proceeded with those settings.

**Step 4**: The BLAST result list from either the NCBI or UniProt databases was manually reviewed to identify "subject" sequences with conserved N-terminal histidines. For NCBI, this was done using the alignment view tab, while in UniProt, the alignment view was set to "wrapped". Each "subject" result was analysed individually, without

ranking them based on their identity or positive percentages, and without considering minimal e-values. In other words, all retrieved "subject" sequences were assessed one by one to check for a histidine aligned at the same position as in the "query" SP + H"X". In UniProt, results from the same taxonomic species as the "query" SP were excluded. Once an amino acid sequence was found with a histidine as the putative N-terminal residue, further analysis was conducted to search for additional amino acids capable of coordinating with metals, such as another histidine (as in a histidine brace motif) or other residues capable of coordinating a metal ion, where sequences which had several potential conserved amino acids that could coordinate given higher consideration. Any sequences which had the second amino acid in position 2 (i.e., immediately after the N-terminal histidine) were rejected as steric constraints would make simultaneous coordination by the side chains of both amino acids highly unlikely.

**Step 5**: After confirming the presence of a second amino acid residue capable of coordinating with metal in the retrieved sequence, the sequence was subjected to signal peptide analysis as described in Step 1. Once histidine was confirmed as the N-terminal amino acid following the SP of the target sequence, the entire sequence was used as the query and BLASTed against the NCBI or UniProt databases, as described in Step 3, for homologous/orthologous searches (refer to Supplementary Data 1–4 for the list of retrieved sequences). These sequences were then processed through the pipeline outlined below for protein domain/family identification.

### In-silico analysis of the target sequences

Amino acid sequences retrieved from the signal strapping method were submitted to a search, in default mode, at the InterPro database (https://www.ebi.ac.uk/interpro/)[37] to determine whether the sequences belong to a known protein family (in Pfam, Panther, etc.) or not. After the search, the protein family signature was recovered from the respective Pfam database entry and the probability that histidine is one of the first amino acids in the family signature was determined. For the protein families without an entry in the InterPro/Pfam database, the signature was generated in the ConservFold: Protein conservation to 3D structure generator (https://colab.research.google.com/drive/1s7N6w2VEjadkJVS9bFzyOLaFZ3uImS0k)[38] in default mode, using the amino acid sequence as query. Afterwards, the probability of histidine as the first amino acid from the generated family signature was counted using the logo data file. For the taxonomic classification of the protein family sequences, the sunburst chart was accessed on the Taxonomy tab from the Pfam entry page at the InterPro/Pfam server. The taxonomic determination of the protein families without an entry in the InterPro/Pfam was determined in the Enzyme Function Initiative (EFI) server (https://efi.igb.illinois.edu)[39], using the Taxonomy classification after running the EFI Enzyme Similarity Tool (EFI-EST)[17] to generate sequence similarity networks (SSNs) (https://efi.igb.illinois.edu/efi-est/). The sequence was searched using the fasta mode against the UniProt (with default parameters), generating the sunburst chart.

To explore the genome neighbourhoods from the target protein families, SSN clusters were generated to facilitate the assignment of function within protein families and superfamilies. For this, the amino acid sequence retrieved from the signal strapping method was used to generate an SSN at the EFI server using the Enzyme Similarity Tool (EST) (https://efi.igb.illinois.edu/efi-est/)[17]. The amino acid sequence was blasted against the UniProt database, with default parameters. After SSN generation, the data were transferred to the EFI Genome Neighbourhood Tool (EFI-GNT)[17], and the Neighbourhood Size set to 10, and the Minimal Co-occurrence Percentage Lower Limit to 20, before running the analysis to generate the neighbourhood diagrams. To count the functions of the neighbourhood genes, the diagrams were exported to Gene Graphics and the file was copied in Excel and the "Function" column was copied and pasted to a new spreadsheet, where the Excel Analyse Data function was used to sum the counts of

functions and to determine how many copies of each Pfam or InterPro family were present in the neighbourhood of the target protein family. The families with more the top ten highest counts of function were studied used to get insights to inform the assignment of function within the target protein families.

For the determination of the operon/gene cluster (gene neighbourhood GN) in the genome of the target protein chosen for recombinant production, its genome was accessed at the NCBI database, and the gene ID was used for spatial localisation of the gene in the genome web browser viewer in the graphic tab of the genome webpage in the NCBI. After finding the target gene, its neighbourhoods were manually annotated, showing the four genes upstream and downstream of the target gene.

### Recombinant production and purification of proteins

The CDS of the DUF4198 domain (WP_082368692.1) from *Ideonella sakaiensis* (IDE_SAK), excluding the native signal peptide, was synthesised by GenScript. Overhang sequences were added to the DUF4198 gene to facilitate sub-cloning into the pET SUMO vector using the In-Fusion HD Cloning Kit (Takara). The pET-SUMO-*Is*DUF4198 construct was used to transform *E. coli* T7 Shuffle competent cells, which were then cultivated in Terrific Broth (TB) medium containing kanamycin at a concentration of 35 μg/mL at 30 °C and 250 rpm. When the optical density reached 0.6, protein expression was induced by adding 0.5 mM IPTG. Protein expression was conducted for 16 hours at 16 °C and 180 rpm. The cells were pelleted by centrifugation at $10,000 \times g$ for 30 min and resuspended in buffer A (20 mM sodium phosphate, pH 7.4; 500 mM NaCl; 5 mM imidazole) containing lysozyme (0.05 mg/mL) and a cOmplete protease inhibitor cocktail (Roche). The resuspended cells were sonicated, and the cell debris was removed by further centrifugation at $38,000 \times g$ for 20 min using a Lynx Sorvall centrifuge.

The supernatant was loaded onto a HisTrap column 5 mL (GE Healthcare) pre-equilibrated with buffer A (20 mM potassium phosphate pH, 7.4; 500 mM NaCl; 5 mM imidazole). Bound proteins were eluted using a gradient of 0 – 100% of Buffer B (20 mM potassium phosphate, pH, 7.4; 500 mM NaCl; 500 mM imidazole). Fractions containing 6xHis-Sumo-*Is*DUF4198 were combined and diluted in a solution containing 20 mM potassium phosphate, pH, 7.4; 500 mM NaCl to lower the imidazole concentration. Subsequently, 10 μg of SUMO protease per mg of 6xHis-Sumo-IsDUF4198 was added to the solution, which was incubated for 16 hours at 4 °C. The protein was then loaded onto a His-Trap column 5 mL, and the flow-through was collected. *Is*DUF4198 was concentrated with Vivaspin concentrator (cut-off 10 kDa, Sartorius) and loaded onto a HiLoad 16/600 Superdex 75 gel filtration (GF) column (GE Healthcare), equilibrated with GF buffer (20 mM sodium phosphate, pH 7.4; 150 mM NaCl). The fractions containing *Is*DUF4198 were combined and stored at -20 °C until further use. After identifying the correct metal binding to the protein structure, *Is*DUF4198 was incubated with a 2-fold excess of $NiCl_2$. The protein was then passed over the gel filtration column again under the same conditions to remove any unbound nickel ions. Protein concentration was estimated by measuring absorbance at 280 nm with a NanoDrop spectrophotometer, using the molecular weight and extinction coefficient of the target protein construct.

A CDS for a DUF6702 protein from *Pseudomonas aeruginosa* (VZT40374.1) was used to design a construct which lacked its native signal peptide and putative C-terminal domains, and which had a C-terminal Strep-tag® II (WSHPQFEK) followed by a stop codon added, and this was synthesised by Genscript. The CDS construct (*Pa*DUF6702 + Stre-tagII) was cloned into a pET-26b(+) expression plasmid between the pelB signal sequence and BamHI restriction site using Gibson assembly by Genscript. The expression plasmid (ca 50 ng) containing the *Pa*DUF6702+Stre-tagII gene was then transformed by heat shock into *E. coli* (DE3) Rosetta 2 *pLysS*, plated on LB agar

containing 35 µg/mL of kanamycin and 34 µg/mL of chloramphenicol and incubated ON at 37 °C. For recombinant protein expression, a single transformed colony was inoculated into 20 mL Terrific Broth (TB) with the antibiotics and cultivated at 37 °C and 210 RPM for 16 h. 15 mL of this culture was inoculated into 1 L of M9(+)[40] with the antibiotics and incubated at 37 °C and 250 RPM, until it reach an optical density (OD) of 5.0. The temperature was reduced to 16 °C, and the culture was acclimated for 1 h, and then protein expression was induced with 1 mM IPTG for 24 h.

After expression, the cells were harvested at $10,000 \times g$ for 30 min and resuspended in ice-cold binding buffer (100 mM HEPES, pH 8.0 with 20% [w/v] sucrose) and left on ice for 30 min with small agitation before a further round of centrifugation at $28,000 \, g$ for 20 min. The supernatant was kept, and the pellet was resuspended in ice-cold 5 mM MgSO4 and left on ice for another 30 min. Following a second centrifugation at $28,000 \times g$ for 20 min, the supernatant was collected and combined with the previous supernatant and its pH adjusted to 8 as necessary, using 1 M HEPES pH 8.0 buffer. The periplasmic extract was then loaded at 0.5 mL/min onto a 5 ml StrepTrap HP column (Cytiva), equilibrated with binding buffer, and the protein was eluted with 50 mM HEPES pH 8.0, containing 300 mM NaCl and 2.5 mM desthiobiotin. Protein concentration was determined by measuring absorbance at 280 nm with a NanoDrop spectrophotometer, using the molecular weight and extinction coefficient of the strep-tagged protein. The *apo* form *Pa*DUF6702 was concentrated using a VivaSpin 5 kDa concentrating device and then treated with 20 mM EDTA to remove any metals present, and finally, the protein was concentrated to 5 ml and 1 mL aliquots were subjected to further purification by GF on a HiLoad 16/600 Superdex 75 GF column (Cytiva) equilibrated with 50 mM Tris-HCl buffer pH 8.0 containing 200 mM NaCl. After identifying the correct metal to bind to *Pa*DUF6702, the protein was incubated with a 2-fold excess of metal ON and the protein run down the gel filtration column again.

The coding sequence (CDS) of the *Rb*Ang-1 domain from *Rhodobacteraceae bacterium* PD-2 (ETA49561.1), excluding the native signal peptide and the HupE/UreJ 2 C-terminal domain, was fused to an N-terminal 6xHis-tag-SUMO protein gene immediately upstream of the His1 codon of *Rb*Ang-1. A C-terminal Strep-tag® II (WSHPQFEK) was also added to the synthetic CDS, followed by a stop codon. The construct was synthesised by GenScript and subcloned into pET-28a(+) between the NcoI and TaqI restriction sites. The expression plasmid (~10 ng) containing the 6xHis-tag-SUMO-*Rb*Ang-1a-StreptagII gene was then transformed by heat shock into *E. coli* B Shuffle *pLys Y* T7 Express, plated on LB agar containing 35 µg/mL of kanamycin and incubated at 37 °C overnight (ON). For recombinant protein expression, a single transformed colony was inoculated into 50 mL LB containing 35 µg/mL of kanamycin and cultivated at 37 °C and 180 RPM. After ~ 12 h, 10 mL was inoculated into 1 L of LB with the antibiotic and incubated at 37 °C and 180 RPM, until reach an OD of 0.8 – 1.0. The temperature was then reduced to 18 °C, and the culture was acclimated for one hour and the protein expression was induced with a final concentration of 0.5 mM isopropyl β-D-1-thiogalactopyranoside (IPTG) for 16 h.

After harvesting the cells by centrifuging at $10,000 \times g$ for 30 min, the cell pellets were resuspended in five times their volume of buffer A (50 mM HEPES, pH 8; 150 mM NaCl; 30 mM imidazole). The resuspended cells were then sonicated, and the cell debris was removed by further centrifugation using a Lynx Sorvall at $38,000 \times g$ for 20 min. Next, the supernatant was loaded onto a HisTrap crude 5 ml Ni column (GE Healthcare) pre-equilibrated with buffer A. The column was washed with 5 column volumes (CVs) of buffer A and a gradient of 0 – 100% buffer B (50 mM HEPES pH 8; 150 mM NaCl; 300 mM imidazole) applied over 20CVs. The fractions containing the 6xHis-tag-SUMO-*Rb*Ang-1a -Streptag-II protein were combined, concentrated using a VivaSpin 10 kDa cut-off concentrating device, and then diluted tenfold with buffer A to lower the imidazole concentration. The protein concentration was determined by measuring the absorbance at 280 nm with a NanoDrop spectrophotometer and using the molecular weight and extinction coefficient of the target protein construct. DTT was added to a concentration of 1 mM, along with 10 µg of SUMO protease per mg of 6xHis-tag-SUMO- *Rb*Ang-1a -Streptag-II. The mixture was incubated overnight at 20 °C with shaking at 10 rpm. The protein solution was then passed through another crude 5 ml Ni column equilibrated with buffer A, and the flow-through was collected. This flow-through was treated with 20 mM EDTA to remove any metals present, and finally, the protein was concentrated to 5 ml and 1 mL aliquots were loaded onto a HiLoad 16/600 Superdex 75 (GE Healthcare) gel filtration (GF) column, equilibrated with 50 mM MES pH 6; 150 mM NaCl. The peak fractions containing the *apo* form *Rb*Ang-1a-StreptagII were combined and then incubated with a 2-fold excess of NiCl2, with further gel filtration to remove any unbound nickel ions again under the conditions mentioned above. Since the Ni-loaded protein was precipitating, for biochemical and crystallisation purposes, the protein was treated with 5 mM EDTA and passed down the gel filtration column again under the conditions above. The fractions were concentrated, a 2-fold excess of CuCl2 was added, and the protein run down the gel filtration column again.

The CDS of the Ang-2 domain from *Microbacterium maritypicum* MF109 (T5KLJ9), excluding the native signal peptide and the HupE/UreJ 2 C-terminal domain, was fused to an N-terminal 6xHis-tag-SUMO protein gene immediately upstream of the His1 codon of *Mm*Ang-2a. A C-terminal Strep-tag® II (WSHPQFEK) was also added to the synthetic CDS, followed by a stop codon. The construct was synthesised by GenScript and subcloned into pET-28a(+) between the NcoI and TaqI restriction sites. The expression plasmid (~10 ng) containing the 6xHis-tag-SUMO-*Mm*Ang-2a-StreptagII gene was then transformed by heat shock into *E. coli* B Shuffle *pLys Y* T7 Express, plated on LB agar containing 35 µg/mL of kanamycin and incubated at 37 °C for 16 h. For recombinant protein expression, a single transformed colony was inoculated into 50 mL LB containing kanamycin (35 µg/mL) and cultivated at 37 °C, 180 RPM. After ~12 h, 10 mL was inoculated into 1 L of LB and incubated at 37 °C and 180 RPM, until an OD of 0.8 – 1.0. The temperature was then reduced to18 °C, and the culture was acclimated for one hour and the protein expression was induced with a final concentration of 0.5 mM IPTG for 16 h.

After harvesting the cells by centrifuging at $10,000 \times g$ for 30 min, the cell pellets were resuspended in five times their volume of buffer A (50 mM HEPES, pH 8; 150 mM NaCl; 30 mM imidazole), sonicated, and the cell debris was removed by further centrifugation using a Lynx Sorvall at $38,000 \times g$ for 20 min. The supernatant was loaded onto a HisTrap crude Ni column 5 mL (GE Healthcare) pre-equilibrated with buffer A, and then eluted using a gradient of buffer B (50 mM HEPES pH 8; 150 mM NaCl; 300 mM imidazole). Fractions containing the 6xHis-tag-SUMO-*Mm*Ang-2a-StreptagII protein were concentrated using VivaSpin 10 kDa cut-off, and then diluted tenfold with buffer A to lower the imidazole concentration. Protein concentration was determined by measuring absorbance at 280 nm with a NanoDrop spectrophotometer, using the molecular weight and extinction coefficient of the target protein construct. Subsequently, 10 µg of SUMO protease per mg of 6xHis-tag-SUMO-*Mm*Ang-2a-StreptagII was added to the solution, along with DTT (1 mM final concentration), which was incubated for 16 h at 20 °C. The protein was then loaded onto a HisTrap column 5 mL, and the flow-through was collected. This was treated with 20 mM EDTA to remove any metals present, and finally, the protein was loaded onto a HiLoad 16/600 Superdex 75 (GE Healthcare) GF column, equilibrated with 50 mM MES, pH 6; 150 mM NaCl. The peak fractions containing the *apo* form *Mm*Ang-2a -StreptagII were combined and concentrated for analysis. *Mm*Ang-2a-StreptagII was incubated with a 2-fold excess of CuCl2, then the protein was gel filtrated again

under the conditions mentioned above to wash out any unbound copper ions.

## Thermal shift assay (TSA) for Metal Bind Screening

*Apo*-proteins were prepared prior TSA screening by incubating 200 μL of target protein at 5 mg/mL with 20 mM EDTA for, at least 16 hours in the fridge (4–6 °C), after which the EDTA and EDTA complex were removed through size-exclusion chromatography, using a Superdex-75 10–300 GL on its appropriate buffer as reported above. In addition, immediately prior to TSA measurements, one protein sample at 15 μM was treated again with 5 mM EDTA and subjected to a TSA assay to confirm a lack of any significant thermal shift difference between the apo form and the EDTA-treated enzyme.

The TSA was conducted to determine the temperature of melting ($Tm$) of the purified *apo* target proteins in the presence of several metals. The method was carried out with SYPRO® Orange Protein Gel Stain (Life Technologies) as the probe, using an Mx3005P qPCR System (Agilent Technologies). For this, solutions of the *apo*-proteins at 30 μM were incubated with different ratios of metal: protein molar equivalents, such as 200:1, 10:1 and 1:1. The incubation of the protein with the metal chloride solutions $MnCl_2$, $CoCl_2$, $FeCl_2$, $NiCl_2$, $MgCl_2$, $ZnCl_2$, $CuCl_2$, and $CaCl_2$ was performed for 3 h on ice, with occasional mixing after every 30 minutes. Subsequently, 25 μL of this solution was mixed with 5 μL of diluted SYPRO® Orange (1:99 of $H_2O$) and loaded in PCR tube strips with flat optically clear caps. The intensity of the fluorescence was measured over a temperature gradient of 20–95 °C (30 sec hold on each temperature) and converted into a melting curve (fluorescence changes against temperature) by the JTSA[41] website https://paulsbond.co.uk/jtsa/#/input to determine the $T_m$, using a sigmoid 5 curve model function and inflection to determine the melting point[42].

## TSA for determine dissociation and binding constants

The TSA was conducted to determine the temperature of melting ($Tm$) of the 20 mM EDTA-treated and gel-filtrated *apo* target proteins in the presence of several metal concentrations. 100 μL samples of 5 μM apo form targeted proteins in their respective size-exclusion buffer were incubated with several metals' concentrations varying from 0–100 μM during 3 h on ice with some mixing every 30 min. Subsequently, 25 μL of this solution was mixed with 5 μL of diluted SYPRO® Orange (1:99 of $H_2O$) and loaded in PCR tube strips with flat optically clear caps. The intensity of the fluorescence was measured over a temperature gradient of 20–95 °C (30 sec hold on each temperature) using an Mx3005P qPCR System (Agilent Technologies). After data collection, columnar txt files were submitted to the Thermott (https://thermott.com) webserver[43] to determine the melting temperature of the protein ($T_m$), ligand affinity ($K_b$) calculation, odd the dissociation constants ($K_b$). The analysis was performed as reported in Thermott's documentation guidance. Briefly, after file uploading, the melting temperature of each sample was manually fitted using the thermodynamic preset. The $K_b$ and $K_d$ were determined after loading the experiment builder with the protein number of amino acids and binding temperature set to 25 °C as parameters. The other parameters were applied as default. Fitting analyses were performed automatically by the web server with manual adjustment of melting temperature when necessary. Plots containing the binding and dissociation curves were exported from the web server.

## Protein crystallisation, X-ray data collection, structure solution and refinement

Pure metal-loaded target proteins in their respective GF buffers were screened in several commercial crystallisation screens using the sitting drop vapour diffusion technique, and using a Mosquito robot (TTP Labtech, Melbourn, UK) to set up the plates. The commercial kits

utilised were: JCSG + (Molecular Dimensions), PACT Premier (Molecular Dimensions), Crystal Screen HT (Hampton Research), Peg Ion HT (Hampton Research), Index (Hampton Research), Salt Rx (Hampton Research), and XP Screen (Jena Bioscience).

*Is*DUF4198 was buffer exchanged into 20 mM Tris pH 7.5 at 30 mg/ml, and Ni chloride was added to 2 mM. The protein was crystallised using the JCSG + screen, condition F10 (1.1 M Na malonate, 0.5% v/v Jeffamine Ed-2001, 0.1 M HEPES pH 7). A stock of seed crystals was grown against a reservoir solution comprised of 0.8 M Na malonate, 0.1 M HEPES, pH 7.0 and 0.5 % (w/v) Jeffamine D2001, using a ratio of 0.35 μL protein: 0.5 μL reservoir solution. Larger crystals were grown using a reservoir solution consisting of 1.1 Na malonate, 0.1 M HEPES pH 7.0 and 0.5 % (w/v) Jeffamine D2001, using a ratio of 0.3 μL protein: 0.1 μL seed: 0.4 μL reservoir solution. A crystal was fished into liquid nitrogen via a cryoprotectant solution comprised of 3 M Na malonate, 0.1 M HEPES pH 7.0, 0.5 % Jeffamine D2001.

Data were collected at the Diamond Light Source on beamline io3 at a wavelength of 0.97950 and processed with DIALS[44,45]. CCP4cloud was used for further processing[46]. *AIMLESS* was used for data reduction to a resolution 1.44 Å[47]. The structure was solved by molecular replacement with *Phaser*[52] using a model from AlphaFold2 and refined with repeated cycles of *REFMAC5*[48] and model building in *Coot*[49]. Structure figures were made using Pymol[50].

Pure copper-loaded *Rb*Ang-1a (40 mg/mL) in 50 mM MES pH 6.0, 150 mM NaCl was crystallised using the Hampton Research Crystal Screen HT condition G3 (10 mM Zn sulfate, 0.1 M MES pH 6.5 and 25 % (v/v) polyethylene glycol monomethyl ether 550). The crystals were fished without further cryoprotection into liquid nitrogen. Diffraction data (Supplementary Table 1) were collected at the Diamond Light Source (UK), beamline io3, at a wavelength of 0.97628 Å. Diffraction data were indexed, integrated and scaled with XDS[51]. Further data processing was performed in CCP4cloud[46], using *AIMLESS*[47] for data reduction at 2.25 Å resolution. The structure was solved by molecular replacement with *Phaser*[52] using a model generated by AlphaFold2[31], and further model building and refinement steps were performed with cycles of *Coot* and *REFMAC5*[48,49]. An anomalous map confirmed the position of copper in the histidine brace as a positive density peak. Three further anomalous peaks were modelled with zinc ions, which were in the crystallisation condition. It is also possible that these sites may be occupied by copper, but it is not possible to assign the density to either metal ion unequivocally. Structure figures were made using Pymol[50].

## Protein modelling using AlphaFold3 Server

The AlphaFold webserver (https://alphafoldserver.com), powered by AlphaFold3[3], was used (accessed on: June, July and August 2024 and January 2025) to run structural predictions of the Apo- Cu- and Co-forms of *Pa*DUF6702. For this, the sequence of *Pa*DUF6702, lacking the native signal peptide and putative C-terminal domains, was uploaded onto the webserver without ligands, or in the presence of one copy of $Cu^{2+}$ or $Co^{2+}$ as entities (ligands). For *Mm*Ang-2a, a different sequence approach was employed. In this case, only the signal peptide was removed since the HupE/UreJ-2 Pfam domain at the C-terminal of *Mm*Ang-2a was predicted to bind metals. Therefore, the full-length sequence of *Mm*Ang-2a-HupE/UreJ-2 Pfam was loaded in the web server without ligands, and in the presence of two copies of $Cu^{2+}$ as entities. All the predictions were run in automatic seeding mode, and the global pLDDT, pTM and ipTM scores were used as parameters to determine the accuracy of the structural prediction without and with the ligand interfaces.

## Electrostatic surface potential (ESP) and ConSurf analysis

The adaptive Poisson-Boltzmann Solver (APBS)[53] programme was used to calculate the electrostatic potential molecular surface of each target

protein family using the Pymol plugin. Either the crystal structure pdb file or the AF3 model cif file was used as a target for pdb2pqr molecule preparation, with the maps set for calculation using 0.50 of grid space. Molecular surface visualisation was performed with the range set at ± 2 $k_BT/e$ and set as solvent accessible. The pH was set in the advanced configuration tab, with the command line option: --ff = AMBER --with-ph = 7.0 --ph-calc-method = propka. For ConSurf analysis, AF2[31,54] models of each target protein family were generated and submitted to the ConSurf server (https://consurf.tau.ac.il/consurf_index.php) with default parameters[55]. Briefly, HMMER was chosen as a homologue search algorithm, with one interaction and an e-value cutoff of 0.0001, using Uniref-90 as a database. For the selection of homologues, 150 sequences were chosen with sequence identities between a minimum of 35% and a maximum of 95%. MAFFT-L-S-I was used to build the multiple sequence alignment (MSA) with neighbour-joining with ML distance for phylogenetic tree building. Conservation scores were calculated with the Bayesian method, and the amino acid substitution model was chosen by best fit.

### Electron paramagnetic resonance (EPR) spectroscopy analysis and data simulation

Continuous wave (cw) X-band EPR spectra were collected at 150 K for a frozen solution of the target protein with a concentration of ca. 0.25 mM in 50 mM HEPES, pH 7.0 for $Pa$DUF6702 and 50 mM sodium acetate buffer, pH 5 for $Rb$Ang-1a and $Mm$Ang-2a. Data collection was performed using a Bruker micro EMX spectrometer using a frequency of ca. 9.30 GHz, with modulation amplitude of 4 G, modulation frequency of 100 kHz and a microwave power of 10.02 mW for $Pa$DUF6702 and 5.02 mW for $Rb$Ang-1a and $Mm$Ang-2a. The data were intensity-averaged over eight scans. Simulations of the experimental data were performed using the Easyspin 5.2.28[56] open-source toolbox implemented by MATLAB R2020a software on a PC.

### Biochemical assays for metal chemistry

$Apo$-proteins were prepared prior to biochemical assays by incubating them with 20 mM EDTA for at least 16 h, after which the EDTA and EDTA complex were removed through size-exclusion chromatography (as reported above). After, 1 molar equivalent of metal (copper, nickel or cobalt) was further incubated with the target proteins for more 16 h, followed by another round of size-exclusion chromatography. Fractions containing the metal-loaded forms of the proteins were concentrated as before mentioned.

The oxidase activity of the target proteins was assessed using the Amplex® Red assay to quantify $H_2O_2$. This experiment followed the protocol by Kittl et al. (2012)[21]. Triplicate samples were prepared in a clear microplate with a final volume of 100 μL. Each assay included 50 μM of ascorbic acid, 50 μM Amplex® Red, 7 U/mL Horseradish Peroxidase (HRP), and 1 μM target protein in 1 x PBS at pH 7.2. Reactions were initiated by adding the ascorbic acid, and the formation of the product (resorufin) was monitored by measuring absorbance at 570 nm. The kinetics were recorded for either 30 or 60 min at 30 °C using the Epoch 2 Microplate Reader (BioTek), with 3 s of shaking before each reading. The results were reported as apparent hydrogen peroxide production.

The peroxidase activity assay using 2,6-dimethoxyphenol (2,6-DMP) and $H_2O_2$ as co-substrates was conducted as described by Breslmayr et al.[57]. Reactions were prepared in a total volume of 200 μL, with final concentrations of 0.1 mM $H_2O_2$, 10 mM 2,6-DMP, and 1 μM of target proteins in a 100 mM ammonium acetate buffer at pH 6, maintained at 30 °C. The reaction was initiated by adding the protein with the reagents, which had been pre-incubated for 15 min at 30 °C. The absorbance was measured at 469 nm every 15 seconds for 5 or 30 min using the Epoch 2 Microplate Reader (BioTek) with the correction path length option enabled. Peroxidase activity was reported as apparent cerulignone production.

### In vitro activity assays using polysaccharides and product degradation analysis by mass spectrometry

Typical reactions for carbohydrate-active enzyme characterisation involved mixing 5 mg/ml of the substrate with 1 μM of purified target protein and 2 mM ascorbic acid in a total volume of 100 μL. These reactions were carried out in a 20 mM ammonium acetate buffer at pH 6 and incubated for 24 h at 30 °C with shaking at 1000 RPM using a Thermomixer. After top speed centrifugation for 10 min, the supernatant of the reactions was analysed using MALDI-TOF MS. For spotting, 1 μl of supernatant was mixed with an equal volume of 20 mg/ml 2,5-dihydroxybenzoic acid (DHB) in 30% acetonitrile and 0.1% TFA on a SCOUT-MTP 384 target plate (Bruker). The samples were air-dried using a bulb lamp before being analysed by mass spectrometry on an UltrafleXtreme matrix-assisted laser desorption ionisation-time of flight/time of flight (MALDI/TOF-TOF) instrument (Bruker). To obtain high-resolution mass spectra, data were collected by averaging around 10,000 shots from a 2 kHz smartbeam-II laser. MS scans were run in reflector mode across a mass range of 300–5000 Da, followed by MS/MS fragmentation using LIFT-CID with argon as the collision gas, when required[58]. Instrument operation and data processing were handled by FlexControl and Flex-Analysis software, respectively.

### In vitro activity assays for protease activity

Protease activity was assessed using 1% (w/v) Azocoll, Azocasein, and Keratin Azure as substrates. Reactions were conducted in 200 μL volumes containing 1 μg of purified target protein and 1x PBS buffer (pH 7.4). The mixtures were incubated at 37 °C for 24 h at 300 RPM. The enzymatic reactions were stopped by adding 50 μL of 10% (w/v) trichloroacetic acid (TCA) and allowed to stand for 10 min at room temperature to precipitate any undigested substrate. Following centrifugation at 10,000 x $g$ for 5 min, the supernatant was transferred to new cuvettes or microplate wells, and the absorbance was measured at 520 nm for Azocoll, 440 nm for Azocasein samples (after addition of 100 μL 0.5 M NaOH), or 595 nm for Keratin Azure. Protease activity was quantified by comparing sample absorbance to a standard curve generated with known substrate concentrations, with enzyme activity expressed in units defined as the amount of enzyme required to produce an increase in absorbance by 0.01 per minute under the assay conditions.

### Phylogenetic analysis of the target proteins and their homologues

Phylogenetic analysis was conducted on the NGPhylogeny.fr platform[59] using the A la carte mode. Briefly, the target sequences and their homologues (Supplementary Data 1–4) had their signal peptide predicted by SignalP 6.0[16], and then removed, revealing His1 as the initial amino acid in each case. After, the sequences were loaded in the NGPhylogeny webserver, aligned using MAFFT[60] with default settings and without any curation. Maximum likelihood analyses were subsequently carried out to explore the phylogenetic relationships among the herein identified metalloprotein families. The alignment generated by MAFFT was utilised to build the phylogenetic tree via PhyML[61], with default parameters, except for gaps, which were retained. Bootstrap tests (set to 100)[62] were applied to assess branch support, and the phylogenetic tree was visualised using iTOL[63].

### Protein flexibility and protein motion and dynamics simulations

AF3 models of Apo $Rb$Ang-1a-HupE/UreJ-2 and $Mm$Ang-2a-HupE/UreJ-2 were uploaded as pdb files in the Dynamics tool from the Campus Chemical Instrument Centre (CCIC) web server (https://spin.ccic.osu.edu/index.php/dynamics/index) for the prediction of N − H order parameters in globular proteins[64]. After data processing, the linker region between the Anglerase and the HupE/UreJ-2 domains was evaluated for the $S^2$ order parameter for backbone N − H vectors and,

therefore, to predict protein flexibility[65]. For protein motion and dynamics simulations, AF3 models of Apo *Rb*Ang-1a-HupE/UreJ-2 and *Mm*Ang-2a-HupE/UreJ-2 were uploaded as PDB files in the DynaMut web server (https://biosig.lab.uq.edu.au/dynamut/) under the normal mode analysis (NMA)[66] tab, using c-alpha as force field. Vector trajectory representations for the first non-trivial mode of the molecule motion and Porcupine plots from the vector field representation of the molecule for the first non-trivial mode of the molecule were downloaded and analysed in Pymol.

## Reporting summary

Further information on research design is available in the Nature Portfolio Reporting Summary linked to this article.

## Data availability

Crystallographic data for the structures reported in this article have been deposited at the Protein Data Bank (PDB), under accession codes 9GCB for *Is*DUF4198 and 9GCE for *Rb*Ang-1a. The remaining data are available in the main paper figures, Supplementary Information, Supplementary Data, and Supplementary Movie. Any additional information is available upon request to the corresponding authors. Source data are provided in this paper.

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

## Acknowledgements

We would like to thank the Bioscience Technology Facility from the University of York, especially Chris Taylor and Adam Dowle from the Centre for Excellence in Mass Spectrometry and Andrew Leech from the Molecular Interaction Laboratory. We also thank the York Structural Biology Laboratory (YSBL), especially Johan Turkenburg and Sam Hart. AKN holds a Daphne Jackson Trust Fellowship and would like to thank the University of York and the Royal Society of Chemistry for their support in funding the research. This work was funded by the Biotechnology and Biological Sciences Research Council (BBSRC) (BB/V004069/1) and the European Research Council (ERC 951231).

## Author contributions

J.P.L.C. expanded the method, performed and analysed the bioinformatic, molecular biology, biochemical and spectroscopy data and drafted the manuscript. T.L.R.C. performed the bioinformatic, molecular biology, and biochemical data. W.A.O. performed and analysed molecular biology, biochemical and spectroscopy data. A.K.N. performed and analysed spectroscopy data. J.W. performed and analysed molecular biology data. S.T.S. performed and analysed molecular biology data, provided supervision and acquired funding. G.J.D. analysed the whole data, wrote the manuscript, provided supervision and acquired funding. P.H.W. conceived the method, performed bioinformatic data, analysed the whole data, wrote the manuscript, provided supervision and acquired funding. All authors discussed the results and reviewed and approved the final versions of the manuscript.

## Competing interests

The authors declare no competing interests.
