## [Transparent Peer Review file · Nature Communications]

Signal-strapping as a protein-sequence search method for the discovery of metalloproteins

Corresponding Author: Professor Paul Walton

Version 0:

Reviewer comments:

Reviewer #1

(Remarks to the Author)

Cairo et al. present a new signal strapping methodology which uses N-terminal signal peptide sequences are tagged with a histidine residue and then used in proteomic searches. This is used to identify new metalloproteins that have an N-terminal His that can coordinate a metal ion (as in the LPMOs, which the York group have studied extensively). Whilst powerful protein structure prediction tools are currently available, prediction of ligand binding and especially metal ions and metal-cofactors binding lags a long way behind, and development in this area is much needed. The authors demonstrate the use of their signal strapping method to identify 4 new metalloproteins with metal ions bound to the histidine that was tagged onto the signal peptide.

An exhaustive description of the methodological framework is detailed step-by-step. This relies on many bioinformatics tools and online databases: SignalP-6.0 for the identification of signal peptides, NCBI or UniProt for BLAST searches and alignments, the InterPro database or the Enzyme Function Initiative – and their respective servers /tools such as Pfam/Panther or EFI-EST – to identify known or unknown protein families, and the genome neighbourhood tool to infer biological function of the chosen target protein. The methodology is well-described and this will be able to be reproduced, although it is manually operated and may depend on user knowledge and biases in guiding the process.

An impressive and substantial component of the paper is the identification of 4 novel bacterial metalloproteins, and the expression and characterisation of these proteins (including metal binding, EPR spectroscopy and crystal/alphafold structures). In particular, the method identifies two proteins, classed as anglerases, which are proposed as new metallochaperones.

The paper represents a new and different way of identifying metal-binding proteins, and will be of widespread interest. Points to consider.

Lines 67-77 and 79-84: it is not completely clear how the description in lines 67-77 leads directly to the signal strapping technique outlined in 86-99. Can the authors clarify to ensure this is properly understood?

Line 88: The justification for the choice of initial signal peptide sequences is not made clear. To consider adding this to the relevant points in the paper (e.g. for DU4198 line 129 etc).

104: "black dashed bonds" not clear in my copy.

104: Do the authors mean "chemical structure" when they refer to "line diagram"?

140-161: For DUF4198, GN analysis shows proximity to CoB and NikR_C. Also, as noted by the authors, the protein was identified in components for ATP-binding cassette for Ni and Co. The evidence for involvement in both Ni and Co biology seems equally strong, but the protein was designated as a Ni-binding protein based on apparent higher binding affinity in Fig. 2a. But the T_m values are quite similar for Ni, Co and Zn, so how confident are they in the assignment of DUF4198 as a Ni-protein over a Co-protein?

158-161: Isn't the point here that the oxidase activity of the metal-bound protein is weak, so the protein is likely to be just a transporter? If that's the case, it's not clearly phrased (it also mentioned oxidase activity in the main text but oxidase and peroxidase activity in the SI).

167-170: Does the DALI analysis add much to the main message. Ditto for 371-374 and 427-429. The paper is already quite a dense read, so consider moving or shortening.

176: Cylinders are quite hard to see in panel b (and line 407). Also to consider highlighting the Ni atom in panel e.

197: The fact that binding to Cu was also tested should be mentioned earlier (154-160), when discussing binding to Co, Ni, and Zn. Cu seems the only outlier in the series from expected Irving-Williams behaviour.

354: Seems weak evidence to me to designate RbAng-1a as a Ni-protein. Metalation preferences is not entirely determined by binding affinity. Same protein was then crystallised with Cu²⁺ (line 366).

489: gram-positive?

Reviewer #2

(Remarks to the Author)

Reviewer #3

(Remarks to the Author)

The authors of this manuscript present their work on developing a method to identify proteins that contain an N-terminal metal chelation site. These types of chelating sites are found in monooxygenases and copper chaperones. The manuscript uses bioinformatics methods and databases to identify putative metalloproteins with N-terminal binding sites, which are then synthesized using recombinant protein expression and experimentally characterized using thermal shift assays, X-ray crystallography, EPR, and various activity assays. The authors report the characterization of four proteins identified with this method and suggest functions for each protein. While the work presented here provides important contributions to the field of metalloproteins, there are several issues that need to be addressed before publication:

1. Thermal shift assay:

Under the reported assay conditions, each protein was incubated with 5 mM of a metal chloride solution (approximately 200 molar excess). It has been previously reported that the presence of metal ions (even at sub-millimolar concentrations) can destabilize proteins and lead to denaturation and aggregation (doi: 10.1007/s00249-008-0346-4), even in situations where the metal is stabilizing at low concentrations (doi: 10.1107/S1399004714016617). This assay should be performed under different concentrations of metal ions and the presence of protein aggregates should be assessed as they may interfere with the assay results.

Additionally, it is unclear how the apo samples were prepared. The purification methods state that either NiCl₂ or CuCl₂ was used to metalate the proteins. If the apo samples were prepared by skipping this metalation step, the proteins may have still been metalated during expression or purification (especially when Ni columns are used). The absence of a metal bound to the protein in the apo sample should be verified to ensure the correct interpretation of the assay results.

2. Oxidase and peroxidase assays:

The authors conclude that the four proteins characterized are not metalloenzymes based on the oxidase and peroxidase assays. However, while the results are similar to the free metal ions, they appear to be similar to or even surpass the LPMOs tested (Figure S4). Even if the activity is low, not all metalloenzymes would necessarily perform well in these particular assays.

Additionally, based on the aforementioned issues with the thermal shift assay, the most stabilizing metal may not have been the one tested, and the most stabilizing metal may not be the most enzymatically active one. Multiple metals should be tested to more thoroughly explore the potential enzymatic activity of these enzymes before it is concluded that they are not enzymes. Alternatively, more evidence of transport activity could be used to argue that these proteins are transporters and not enzymes.

3. Binding affinities:

The metal ion binding affinities for the proteins should be measured and compared to reported metal transport proteins and metalloenzymes. Binding kinetics may also provide insight into the function of these proteins. The thermal shift assays may be able to provide binding affinities (doi: 10.1038/s41598-018-37072-x).

4. Proposed function of Ang-1 and Ang-2:

While the proposal of an "anglerase" is an interesting idea, there is little experimental evidence provided to support this. While HupE was previously shown to be a Ni transporter, the comparison to the porin for glycan capture is not sufficient as the proposed mechanism is not similar between these two proteins. Not enough information about these anglerase-HupE/UreJ-2 proteins is presented. The proteins may be a part of homo- or hetero-multimeric complex where the flexible linker between the two domains may not allow for the proposed conformational change (AlphaFold-Multimer may give some insight on this possibility). More evidence needs to be provided or a stronger argument made before this proposed function should be accepted.

5. Minor issues:

On line 424 the pLDDT is reported to be ≤ 0.8 , which is a very poor pLDDT score. This is probably misreported metric.

Reviewer #4

(Remarks to the Author)

Reviewer #5

(Remarks to the Author)

The manuscript entitled "Signal-strapping: a protein-sequence search method for the discovery of metalloproteins" reports the development of a method for the discovery of new metalloproteins containing a N-terminal coordinating residue, such as

the histidine found in LPMOs. In particular, "signal strapping" consists in using the sequence of an identified signal peptide with appended a coordinating residue (e.g. histidine) for running a proteomic search. Mature proteins displaying such metal-binding residue at the N-terminal, after cleavage of the signal peptide, are successfully found. Four examples of newly identified metalloproteins have been expressed and characterized, thus demonstrating the applicability of the reported methodology.

In my opinion, the manuscript would be suitable for publication in Nature Communications after addressing some concerns. In particular, given that the "signal strapping" method is the primary focus of the manuscript, I believe that certain aspects of this method need to be described in greater detail. This would provide the reader with all the information required for replicating the procedure, while better illustrating its potential applications.

Following are some suggestions that would help improving clarity and impact of the manuscript:

1) The first step of the signal strapping method is the choice of a known secreted protein, as described on page 3, lines 88-89: "I take a single or consensus sequence of a known secreted protein from an organism (e.g. cellulase) and identify the SP using SignalP 6.0." It is not clear if there is a rationale behind the choice of the starting protein or it is simply a random selection. Could the authors provide an explanation for their choice?

Further, given that the method aims at finding yet undiscovered metalloproteins with a N-terminal coordinating residue (histidine in the examples provided), why did not the authors try using a known LPMO protein as the starting point for their search?

2) I understand that in step 3 of the method pipeline (BLAST and UNIPROT search using the "SP+HX" sequence as a query), a large number of sequences are found (since 500 is put as the maximum number). Can you give an idea of how many target sequences are found at this point, and if all of them are analysed in the subsequent steps?

3) Step 4 of the method is the "manual" review of these sequences for finding a second coordinating residue. This is described at page 17-18, lines 583-585 (experimental section) as follows: "Once a novel amino acid sequence was found with a histidine as the putative N-terminal residue of a new protein, further analysis was conducted to search for additional amino acids capable of coordinating with metals, such as another histidine (as in a histidine brace motif) or aspartic/glutamic acid residues."

Are any specific criteria used for the selection? Which residues, other than histidine, are included? Is the number of coordinating residues or their position in the sequence considered? Are the sequences given a "score" and sorted? I expect that a sequence containing more than one internal histidine would have higher probability to have at least one of them involved in metal coordination with respect to a sequence containing just one glutamine. This aspect should be clarified.

4) Is there any correspondence between the identified internal coordinating residues (apart from the N-terminal His) and those actually found as metal ligands in the experimental structures? For example, Figure 5 shows that the metal binding site of RbAng-1a involves Asn41 as the internal ligand. Was this expected from the selection, or the sequence was selected because it contains other residues (e.g. histidine) that are not found at the metal binding site?

Additional comments:

o The use of terms as "strapping" or "bootstrap" may not be readily clear to non-specialized audience. I believe that their meaning and context should be explained at their first appearance in the text.

o The "signal strapping procedure" is detailed in the introduction section, where also some results are shown regarding the occurrence of N-terminal amino acids in mature vs non-secreted proteins (as reported in figure 1, panels d-f). I believe that this part would be more appropriately placed at the beginning of results and discussion. Accordingly, I would split figure 1, leaving panels 1 a-c in the introduction and panels d-f as a separate figure in the results section.

o Figures in the supporting information should be all recalled in the main text and in sequential order.

Reviewer #6

(Remarks to the Author)

Version 1:

Reviewer comments:

Reviewer #1

(Remarks to the Author)

The authors have made very extensive changes to all of the comments raised by the reviewers, including substantial new data.

Reviewer #2

(Remarks to the Author)

Reviewer #3

(Remarks to the Author)

We thank the authors for properly answering all the questions. It seems good right now.

Reviewer #4

(Remarks to the Author)

Reviewer #5

(Remarks to the Author)

I have really appreciated the significant changes the authors made in the manuscript and the detailed and convincing responses they provided to the comments of all reviewers.

In my opinion, the authors have satisfactorily addressed the reviewers' concerns, and this revised manuscript may now be accepted for publication in Nature Communications.

Reviewer #6

(Remarks to the Author)

REVIEWER COMMENTS

Reviewer #1 **RESPONSES IN RED**

Cairo et al. present a new signal strapping methodology which uses N-terminal signal peptide sequences are tagged with a histidine residue and then used in proteomic searches. This is used to identify new metalloproteins that have an N-terminal His that can coordinate a metal ion (as in the LPMOs, which the York group have studied extensively). Whilst powerful protein structure prediction tools are currently available, prediction of ligand binding and especially metal ions and metal-cofactors binding lags a long way behind, and development in this area is much needed. The authors demonstrate the use of their signal strapping method to identify 4 new metalloproteins with metal ions bound to the histidine that was tagged onto the signal peptide.

We thank the reviewer for these and following comments and for his/her/their careful appraisal of the manuscript.

An exhaustive description of the methodological framework is detailed step-by-step. This relies on many bioinformatics tools and online databases: SignalP-6.0 for the identification of signal peptides, NCBI or UniProt for BLAST searches and alignments, the InterPro database or the Enzyme Function Initiative – and their respective servers /tools such as Pfam/Panther or EFI-EST – to identify known or unknown protein families, and the genome neighbourhood tool to infer biological function of the chosen target protein. The methodology is well-described and this will be able to be reproduced, although it is manually operated and may depend on user knowledge and biases in guiding the process.

We note the comment about user bias, this is undoubtedly true. And while we do not think that such biases significantly reduce the power of the method to discover metalloproteins, since we use both human-selected and consensus sequences as search seeds, we have included the statement below in the Materials and Methods section which acknowledges that bias may be a factor which users need to consider when selecting a search sequence.

“The use of a consensus sequence as a search sequence reduces the human bias which may affect selection of the sequence and the metalloproteins discovered from such a sequence.”

An impressive and substantial component of the paper is the identification of 4 novel bacterial metalloproteins, and the expression and characterisation of these proteins (including metal binding, EPR spectroscopy and crystal/alphafold structures). In particular, the method identifies two proteins, classed as anglerases, which are proposed as new metallochaperones.

The paper represents a new and different way of identifying metal-binding proteins, and will be of widespread interest.

We thank the reviewer for this assessment.

Points to consider.

Lines 67-77 and 79-84: it is not completely clear how the description in lines 67-77 leads directly to the signal strapping technique outlined in 86-99. Can the authors clarify to ensure this is properly understood?

We agree with the lack of clarity highlighted. We have now added the following in the same paragraph.

“First, we assessed whether signal-peptide-containing (SP) proteins were more likely to expose a metal-coordinating amino acid at the N-terminus after SP cleavage. In doing so, we hypothesised that cytosolic proteins with a free N-terminal histidine would chelate redox-active metals (e.g. Cu), potentially generating harmful reactive oxygen species via redox-cycling. As such, the expectation is that such proteins would be less prevalent than those that are secreted.”

Line 88: The justification for the choice of initial signal peptide sequences is not made clear. To consider adding this to the relevant points in the paper (e.g. for DU4198 line 129 etc).

Thank you for this helpful comment. As per the reviewer’s suggestion, we have included a rationale for the choices of each signal peptide at the appropriate places in the manuscript.

We have now included the following justifications.

DUF4198

“GH5 proteins are widely secreted across both eukaryotes and prokaryotes, offering broad taxonomic coverage in downstream searches.”

DUF6702

There is already a justification at the start of this section, but we have now expanded it with the underlined text below.

*“Building on the discovery of Ni-containing DUF4198 proteins and their association with metal-dependent biochemistry, we selected the SP of the DUF4198-domain protein from *Hydrogenophaga* sp. IBVHS1 (Table 1) for signal strapping (Table 1). A blastp search (excluding *Hydrogenophaga* sp.) retrieved DUF4198 homologues and several uncharacterised proteins.”*

Ang-1

*“We applied the signal strapping method using the signal peptide (SP) from a nickel superoxide dismutase (Pfam 09055, WP_230780104.1) of *Roseiconus lacunae* (Table 1). Nickel superoxide dismutases are known to coordinate Ni²⁺ via an N-terminal histidine.”*

Ang-2

*“In a separate search, we used the SP of a GH16 family hemicellulase (IPR000757; SUPfam 49899; WP_230780104.1) from *Streptomyces* sp. YIM 130001 (Table 1). In contrast to the known metalloprotein SP used for Ang-1, GH16s are broadly secreted across bacteria and eukaryotes. A BLASTp search (excluding *Streptomyces* sp.) returned GH16 proteins, metalloproteinase inhibitors (I36), cellulases, and again, proteins with a C-terminal HupE/UreJ-2 domain, with N-terminal domains distinct from Ang-1 (hereafter Ang-2). Like Ang-1, Ang-2 displays a non-cytoplasmic domain after the SP, followed by the TM HupE/UreJ-2 at the C-terminal (Fig. S15a), and ConservFold results showed 100%*

conservation of His1 immediately following SP cleavage, along with highly conserved potential metal-coordinating amino acids in other parts of the sequences (Fig. S15b, Supplementary Data 4)."

104: "black dashed bonds" not clear in my copy.

We have deleted 'black dashed bonds', as they weren't there in the original! Apologies and thank you.

104: Do the authors mean "chemical structure" when they refer to "line diagram"?

We have removed the phrase 'line diagram' which is common parlance in inorganic chemistry, but not more widely. Thank you.

140-161: For DUF4198, GN analysis shows proximity to CoB and NikR_C. Also, as noted by the authors, the protein was identified in components for ATP-binding cassette for Ni and Co. The evidence for involvement in both Ni and Co biology seems equally strong, but the protein was designated as a Ni-binding protein based on apparent higher binding affinity in Fig. 2a. But the ΔT_m values are quite similar for Ni, Co and Zn, so how confident are they in the assignment of DUF4198 as a Ni-protein over a Co-protein?

Thank you for this suggestion. In response to a similar query from reviewer 2, we have now performed new thermal shift assays over a range of metal-protein ratios (200:1, 10:1 and 1:1), which confirm that DUF4198 is principally a nickel-binding protein. These assays and their conclusions are included in our response to Reviewer 2, please see below.

158-161: Isn't the point here that the oxidase activity of the metal-bound protein is weak, so the protein is likely to be just a transporter? If that's the case, it's not clearly phrased (it also mentioned oxidase activity in the main text but oxidase and peroxidase activity in the SI).

Agreed, and we have rephrased as follows:

"The potential role for DUF4198 in metal transport rather than as an active metalloenzyme finds further support from assays described herein (Materials and Methods), which showed only weak peroxidase activity of isolated Ni-IsDUF4198 (See supplementary results and Fig. S6)."

167-170: Does the DALI analysis add much to the main message. Ditto for 371-374 and 427-429. The paper is already quite a dense read, so consider moving or shortening.

Thank you.

We have shortened the appropriate parts as follows:

DUF4198.

Deleted "The structure, however, has no N-terminus histidine and shares only 15% identity with a low Z score of 5.1."

Ang-1

Deleted "...but where the structures only share 7% identity with a Z-score of 8.1."

Ang-2

Deleted “(10% identity, Z-score 9.7, RMSD 5.2 Å)”

176: Cylinders are quite hard to see in panel b (and line 407). Also to consider highlighting the Ni atom in panel e.

We have amended Figures 2 and 5 in line with the reviewer's comments.

197: The fact that binding to Cu was also tested should be mentioned earlier (154-160), when discussing binding to Co, Ni, and Zn.

Agreed. We have added the following to the section on melting temperatures of DUF4198.

“...and Cu²⁺ (– 0.2 °C).”

Cu seems the only outlier in the series from expected Irving-Williams behaviour.

Yes, this is a good point and one we had not considered in the original submission. Accordingly we have carried out new Thermoshift assays over a range of metal-protein ratios and also determined the Ni-DUF4198 dissociation constant using the method described in Petrauskas *et al* (*TrAC Trends Anal Chem*, **2024**, *170*, 117417). These confirm that DUF4198 is indeed a Ni protein. As such, we have included the following statement.

“At a 1:1 protein:metal molar ratio, the greatest thermal stabilisation was observed with Ni²⁺ ($\Delta T_m = +7$ °C, Fig. 2a), followed by Zn²⁺ (+4.6 °C), Co²⁺ (+4.0 °C) and Cu²⁺ (–0.2 °C). These relative shifts, save for Cu²⁺, are in accord with those expected from the Irving Williams series. Further TSA assays at multiple stoichiometries (200:1 and 10:1, Fig. S4) and a measured dissociation constant for Ni²⁺ ($K_d = 18 \pm 3$ nM; Fig. S5) confirm nickel binding, consistent with the GN analysis described above. The potential role for DUF4198 in metal transport rather than as an active metalloenzyme finds further support from assays described herein (Materials and Methods), which showed only weak peroxidase activity of isolated Ni-IsDUF4198 (See supplementary results and Fig. S6).”

354: Seems weak evidence to me to designate RbAng-1a as a Ni-protein. Metalation preferences is not entirely determined by binding affinity. Same protein was then crystallised with Cu²⁺ (line 366).

Agreed and thank you. In the light of this reviewer's insight along with those of reviewer 3 we undertook a series of new Thermoshift assays for all the proteins described in our manuscript. We further measured the dissociation constants of selected metal ions for each of the proteins. More details are given below in our response to Reviewer 3, but for RbAng-1 specifically we now observe the following shifts in melting temperatures when we use a 1:1 metal: protein ratio: Mn (–0.4 °C), Co (–0.4 °C), Fe (–0.9 °C), Ni (+1.2 °C), Cu (+2.1 °C), Zn (–0.5 °C), indicating that Ang-1 is principally a Cu-binding protein. This claim is affirmed by the dissociation constants of Ni-Ang1 and Cu-Ang1 which are 210 nM and 74 nM respectively. We also have a Cu-RbAng1 crystal structure along with EPR spectroscopy. In the light of these data we have therefore amended the manuscript as follows:

Replaced...

“The largest melting temperature shift ($\Delta T_m = +6$ °C) occurred upon addition of Ni²⁺, followed by Co²⁺ (+2 °C), Mn²⁺ (+1.5 °C) and Cu²⁺ (+1 °C). Other metals decreased the T_m , such as

Mg²⁺ (−3.3 °C) and Zn²⁺ (−6.8 °C). Accordingly, we designate RbAng-1a as a Ni-metalloprotein, albeit that Co²⁺, Mn²⁺ and Cu²⁺ can also bind.

with...

“TSAs were used to assess metal binding by Ang-1 and Ang-2. At a 1:1 molar ratio, the largest increase in melting temperature ($\Delta T_m = +2.1$ °C) was observed with Cu²⁺, followed by Ni²⁺ (+1.2 °C). Other metals caused modest destabilisation: Fe²⁺ (−0.9 °C), Zn²⁺ (−0.5 °C), Mn²⁺ (−0.4 °C), and Co²⁺ (−0.4 °C) (Fig. 4e, Fig. S18). Based on these results, RbAng-1a is designated a Cu-metalloprotein, though Ni²⁺ binding is also significant. Dissociation constants (Kd) determined by TSA were 73.6 ± 2.3 nM for Cu²⁺ and 223 ± 20 nM for Ni²⁺ (Fig. S19), suggesting physiological relevance for both ions.”

and deleted...

“...(the Ni-bound protein was susceptible to precipitation).” From the crystallography section.

In the light of the above discussion, we have also deleted the following:

“Establishing the identity of the metal ion associated with the biochemical function of each is, however, more difficult. While RbAng-1a-Ni²⁺ is more stable than with other metal ions, the same cannot be said to be true of MmAng-2a, which is only marginally more stable with Cu²⁺ than Mn²⁺ and Zn²⁺.”

489: gram-positive?

It is Gram-negative. The HupE/UreJ-2 domain attached to RbAng-1 is shown by DeepLock analysis to be localised in the cytosolic membrane of Gram-negative bacteria.

Reviewer #2 (Remarks to the Author):

Thank you for the time and effort you put into refereeing. It is most appreciated.

Reviewer #3 (**RESPONSES IN BLUE**)

The authors of this manuscript present their work on developing a method to identify proteins that contain an N-terminal metal chelation site. These types of chelating sites are found in monooxygenases and copper chaperones. The manuscript uses bioinformatics methods and databases to identify putative metalloproteins with N-terminal binding sites, which are then synthesized using recombinant protein expression and experimentally characterized using thermal shift assays, X-ray crystallography, EPR, and various activity assays. The authors report the characterization of four proteins identified with this method and suggest functions for each protein. While the work presented here provides important contributions to the field of metalloproteins, there are several issues that need to be addressed before publication:

1. Thermal shift assay:

Under the reported assay conditions, each protein was incubated with 5 mM of a metal chloride solution (approximately 200 molar excess). It has been previously reported that the presence of metal ions (even at sub-millimolar concentrations) can destabilize proteins and lead to denaturation and aggregation (doi: 10.1007/s00249-008-0346-4), even in situations where the metal is stabilizing at low concentrations (doi: 10.1107/S1399004714016617). This assay should be performed under different concentrations of metal ions and the presence of protein aggregates should be assessed as they may interfere with the assay results.

Thank you. We agree and have undertaken new Thermoshift experiments for a range of metal ions for all of the proteins at metal ion to protein ratios of 10:1 and 1:1 (in addition to original experiments done at 200:1. The data for 1:1 metal: protein ratios are shown below.

Shifts in melting temperatures ($\Delta T / ^\circ\text{C}$). **Bold** values are positive shifts which are greater than the average shift value + 1 sd.

DUF4198

Metal ion	Protein: metal ion ratio, shifts in melting temp		
	200:1 $\Delta T / ^\circ\text{C}$	10:1 $\Delta T / ^\circ\text{C}$	1:1 $\Delta T / ^\circ\text{C}$
Mg ²⁺		-0.2	-0.2
Ca ²⁺	-0.2	-0.4	+0.1
Mn ²⁺	-0.1	-0.1	+0.1
Fe ²⁺		+1.2	0
Co ²⁺	+1.9	+4.6	+4.0
Ni ²⁺	+4.2	+7.2	+7.0
Cu ²⁺	-1.0	-5.2	-0.2
Zn ²⁺	+1.3	+2.1	+4.6

DUF6702

Metal ion	Protein: metal ion ratio, shifts in melting temp		
	200:1 $\Delta T / ^\circ\text{C}$	10:1 $\Delta T / ^\circ\text{C}$	1:1 $\Delta T / ^\circ\text{C}$
Mg ²⁺	-0.2		-0.3
Ca ²⁺		-0.7	-0.4
Mn ²⁺	+1.1	-1.0	-0.1
Fe ²⁺	+5.2	-0.4	-0.2
Co ²⁺	+10.8	+9.8	+8.9

Ni ²⁺	+10.3	+11.1	+17.2
Cu ²⁺	+11.0	+8.6	+7.0
Zn ²⁺	-6.25	Nd	-13.4

Ang-1

Metal ion	Protein: metal ion ratio, shifts in melting temp		
	200:1 $\Delta T/ ^\circ\text{C}$	10:1 $\Delta T/ ^\circ\text{C}$	1:1 $\Delta T/ ^\circ\text{C}$
Mg ²⁺			-0.3
Ca ²⁺			-0.4
Mn ²⁺	+1.7	+0.8	-0.4
Fe ²⁺	-1.3	-0.2	-0.9
Co ²⁺	+2.1	+1.2	-0.4
Ni ²⁺	+6.0	+2.8	+1.2
Cu ²⁺	+1.1	+0.6	+2.1
Zn ²⁺	-6.5	-1.3	-0.5

Ang-2

Metal ion	Protein: metal ion ratio, shifts in melting temp		
	200:1 $\Delta T/ ^\circ\text{C}$	10:1 $\Delta T/ ^\circ\text{C}$	1:1 $\Delta T/ ^\circ\text{C}$
Mg ²⁺			
Ca ²⁺	-0.1	-0.3	+1.0
Mn ²⁺	+0.5	-0.3	+0.8
Fe ²⁺	-13.2	-0.5	+1.4
Co ²⁺	-1.6	-2.1	+0.9
Ni ²⁺	-2.8	-4.7	+0.2
Cu ²⁺	+3.0	+3.1	+2.8
Zn ²⁺	+0.1	-3.6	+0.2

Dissociation constants:

Ni-DUF4198, 16(1) nM

Co-DUF6702, 3.7(2) nM. Ni-DUF6702, 5.60(8) pM. Cu-DUF6702, 3.26(9) nM

Ni-Ang1, 220(16) nM. Cu-Ang1, 74(2) nM

Cu-Ang2, 63(3) nM

The conclusions from these data are that DUF4198 is a Ni-protein, DUF6702 is principally a Ni-protein but can also bind Co and Cu strongly, Ang1 is principally a Cu protein, but can also bind Ni moderately strongly, and that Ang-2 is a copper protein.

Our original manuscript concluded that DUF4198 is a Ni-protein, DUF6702 is a non-specific sequesterases, Ang1 is a principally Ni-protein that can also bind Co, Ni and Cu, and that Ang-2 is a Cu protein, with weak binding to Mn and Zn.

As such, the two principal differences between the two sets of conclusions are:

DUF6702 is a sequesterases but one which binds Co and Cu strongly and Ni particularly strongly. Ang1 is not principally a Ni-metalloprotein but principally a Cu-metalloprotein that can also bind Ni moderately strongly.

In this light we have made the following changes to the manuscript:

We have replaced all Thermoshift data in the main text of the manuscript for experiments performed at 1:1 metal:protein stoichiometry. Specifically these are figures 2a, 3a, 4e,f.

Thermoshift data for all metal:protein ratios are now given in the Supplementary Information, specifically Fig. S4, S10, S18 and S20.

For DUF4198, we have added the dissociation constant (see response to reviewer 1, above)

For DUF6702, we have made the following change (including added dissociation constants):

Replaced..

“The recombinant protein was assessed for metal-binding capacity using Thermofluor. The largest shift ($\Delta T_m = +11.0$ °C) occurred upon addition of Cu^{2+} , but this was closely followed by Co^{2+} (+10.7 °C), Ni^{2+} (+10.3 °C) and Fe^{2+} (+5.1 °C) (Fig. 3a). PaDUF6702 is therefore designated as a non-specific metal-binding protein, pointing to a broad range of metal binding in vivo, in accordance with the different metal ions that are associated with its gene neighbourhood proteins.”

with...

*“Given its likely metalloprotein nature, we selected a DUF6702 protein from pathogenic bacterium *Pseudomonas aeruginosa* (VZT40374; hereafter PaDUF6702) for recombinant expression in *E. coli* (Fig. S9). The neighbouring genes are depicted in Fig. S8b. TSA analysis at a 1:1 molar ratio revealed a large stabilising shift upon Ni^{2+} binding ($\Delta T_m = +17.2$ °C), with notable shifts also observed for Co^{2+} (+10.7 °C) and Cu^{2+} (+7.0 °C) (Fig. 3a, Fig. S10). Dissociation constants (Fig. S11) confirmed high affinities: Ni^{2+} (5.60 ± 0.1 pM), Co^{2+} (3.6 ± 0.2 nM), and Cu^{2+} (3.26 ± 0.11 nM). These findings support the designation of PaDUF6702 as a metal sequesterases with a particular affinity for Ni^{2+} . Importantly, oxidase and peroxidase assays showed no significant activity for Ni-, Co-, or Cu-bound forms (Fig. S6), suggesting a non-enzymatic metal-binding or transport function.”*

And replaced...

“Attempts to crystallize the Cu^{2+} -loaded form PaDUF6702 were unsuccessful. Therefore, structure predictions of PaDUF6702 were performed for both apo and metal-loaded forms using the AlphaFold3³ (AF3) server. Five 3D models (models 0 to 4) were generated for each of the apo, Cu-loaded, and Co-loaded forms of PaDUF6702,..”

with...

“Crystallisation of metal-bound PaDUF6702 was unsuccessful. Therefore, structure prediction was performed using AlphaFold3 (AF3). Unfortunately, AF3 is not parameterised for Ni-containing proteins, so as alternatives five models (0–4) were generated for the apo, Cu-loaded, and Co-loaded states. All models exhibited high global pLDDT confidence scores (Supplementary Table S3).”

For Ang-1, we have made the following change (as also partly described above for Reviewer 1, but also including dissociation constants):

“TSAs were used to assess metal binding by Ang-1 and Ang-2. At a 1:1 molar ratio, the largest increase in melting temperature ($\Delta T_m = +2.1$ °C) was observed with Cu^{2+} , followed by Ni^{2+} (+1.2 °C). Other metals caused modest destabilisation: Fe^{2+} (−0.9 °C), Zn^{2+} (−0.5 °C), Mn^{2+} (−0.4 °C), and Co^{2+} (−0.4 °C) (Fig. 4e, Fig. S18). Based on these results, RbAng-1a is designated a Cu-metalloprotein, though Ni^{2+} binding is also significant. Dissociation constants (K_d) determined by TSA were 73.6 ± 2.3 nM for Cu^{2+} and 223 ± 20 nM for Ni^{2+} (Fig. S19), suggesting physiological relevance for both ions.”

For Ang-2

We have replaced:

“MmAng-2a was also assessed for metal-binding capacity (Fig. 2f). The largest melting temperature shift ($\Delta T_m = +3$ °C) occurred upon addition of Cu^{2+} , followed by Mn^{2+} (+0.5 °C), and Zn^{2+} (+0.1 °C) (Fig. 2f). Other metals decreased the T_m , such as Fe^{2+} (−13.20 °C), Mg^{2+} (−4.95 °C) and Ni^{2+} (−2.80 °C). Accordingly, we designate MmAng-2a as a potential Cu-containing metalloprotein, although it is evident that Mn^{2+} and Zn^{2+} can also bind, both of which are commensurate with the metal-dependencies of the GN proteins of MmAng-2a. Furthermore, both RbAng-1a and MmAng-2a displayed modest oxidoreductase activity (Supplementary Information and Fig. S4).”

with...

“MmAng-2a was also assessed for metal-binding capacity (Fig. 2f). The largest thermal shift at 1:1 metal:protein ratio occurred with Cu^{2+} (+1.9 °C), followed by Fe^{2+} (+0.5 °C) (Fig. S20). The dissociation constant for Cu^{2+} was 62.3 ± 2.8 nM (Fig. S21), supporting its classification as a Cu-binding metalloprotein. Fe^{2+} binding may also be relevant, consistent with the metal dependencies of nearby genes in its operon”

and

“Oxidase and peroxidase assays of metal-loaded RbAng-1a and MmAng-2a ($M = \text{Cu}, \text{Co},$ or Ni) revealed no substantial activity relative to LPMOs, though the Cu-loaded forms showed weak oxidase activity (Fig. S6). These levels are comparable to those of free Cu^{2+} and known weak LPMO oxidases, suggesting that the primary function of Ang-1 and Ang-2 is metal binding or transport, though a modest enzymatic role cannot be fully excluded.”

Additionally, it is unclear how the apo samples were prepared. The purification methods state that either NiCl_2 or CuCl_2 was used to metalate the proteins. If the apo samples were prepared by skipping this metalation step, the proteins may have still been metalated during expression or purification (especially when Ni columns are used). The absence of a metal bound to the protein in the apo sample should be verified to ensure the correct interpretation of the assay results.

We apologise for inadvertently omitting this information in the original manuscript. The following has now been added to the Materials and Methods section.

“Apo-proteins were prepared prior TSA screening by incubating 200 μ L of target protein at 5 mg/mL with 20 mM EDTA for, at least 16 hours in fridge (4-6 $^{\circ}$ C), after which the EDTA and EDTA complex were removed through size-exclusion chromatography, using a Superdex-75 10-300 GL on its appropriate buffer as reported above. In addition, immediately prior to TSA measurements, one protein sample at 15 μ M was treated again with 5 mM EDTA and subjected to a TSA assay to confirm a lack of any significant thermal shift difference between the apo form and the EDTA treated enzyme.”

2. Oxidase and peroxidase assays:

The authors conclude that the four proteins characterized are not metalloenzymes based on the oxidase and peroxidase assays. However, while the results are similar to the free metal ions, they appear to be similar to or even surpass the LPMOs tested (Figure S4). Even if the activity is low, not all metalloenzymes would necessarily perform well in these particular assays. Additionally, based on the aforementioned issues with the thermal shift assay, the most stabilizing metal may not have been the one tested, and the most stabilizing metal may not be the most enzymatically active one. Multiple metals should be tested to more thoroughly explore the potential enzymatic activity of these enzymes before it is concluded that they are not enzymes. Alternatively, more evidence of transport activity could be used to argue that these proteins are transporters and not enzymes.

We thank the reviewer for their insightful comment. We have now performed the oxidase and peroxidase assays with Cu(II), Ni(II) or Co(II) for all new proteins reported in our manuscript. The data are shown below and are now included as a revised Figure S6 and caption.

Figure S6 a-c) Oxidase and d-f) peroxidase activities of new metalloproteins with 1:1 ratios of Cu (left), Ni (centre), Co (right). * indicates that metal binding studies associate the protein with tight binding of that metal ion.

We draw the following conclusions from the data in S6.

- 1) The peroxidase activities of all forms of the metalloproteins are significantly less (>~10 fold difference) than that of two known LPMOs. Out of the new proteins which

can be associated with the tight binding of a particular metal ion, Ni-DUF4198 exhibits some weak peroxidase activity. As such, we conclude that there is no strong evidence to support peroxidase activity in any of the new metalloproteins, save potentially for Ni-DUF4198 which may be a weak peroxidase.

- 2) The oxidase activities of the Ni and Co-forms of the new metalloproteins are significantly less (>~10 fold difference) than that of a known LPMO (although we note that LPMOs are known to only be weak oxidases). Of the weak activities seen for Ni-DUF4198 and Co-DUF4198, both are comparable to activity seen for the free metal ion.
- 3) The oxidase activities of the Cu-forms of the new metalloproteins are comparable (<~10 fold difference) to both a known LPMO and free Cu. Of the new proteins which can be associated with the tight binding of Cu, Cu-Ang1 and Cu-Ang2 exhibit weak oxidase activity but only comparable to that of free Cu. As such, we conclude that there is no clear evidence to support strong oxidase activity of any of the new metalloproteins, save for some potential weak activity for the Cu forms of Ang1 and Ang2, albeit indistinct from that of free Cu.
- 4) Notwithstanding the lack of strong evidence of redox activity of any of the new metalloproteins, we acknowledge the reviewer's helpful comments and that some moderation of our original statements about the roles of some of these new metalloproteins is warranted (see below).

We would also like to emphasise here that the principal objective of the paper is to demonstrate the method of signal strapping to discover new proteins and that—moreover—the gene-neighbourhood analyses presented therein are supportive of metal-transport or metal-binding roles.

For DUF4198, we have now included the following statement (also see response to reviewer 1):

“The potential role for DUF4198 in metal transport rather than as an active metalloenzyme finds further support from assays described herein (Materials and Methods), which showed only weak peroxidase activity of isolated Ni-IsDUF4198 (See supplementary results and Fig. S6).”

For DUF6702, we have replaced the following:

“As described below, PaDUF6702 has weak oxidase and peroxidase activities (Fig. S4), but not significant enough compared to non-enzymatic activity of free metal ions to designate it as an oxidoreductase.”

With..

“These findings support the designation of PaDUF6702 as a metal sequesterases with a particular affinity for Ni²⁺. Importantly, oxidase and peroxidase assays showed no significant activity for Ni-, Co-, or Cu-bound forms (Fig. S6), suggesting a non-enzymatic metal-binding or transport function.”

For Ang1 and Ang2

We cannot detect any significant peroxidase activity for M-Ang1 and M-Ang-2 (M = Co, Ni or Cu, Fig S4), and can only see oxidase activity which is the same as the known weak oxidase activity of LPMOs or free Cu.

Accordingly, we have replaced the following...

“Furthermore, both RbAng-1a and MmAng-2a displayed modest oxidoreductase activity (Supplementary Information and Fig. S4).”

With...

“Oxidase and peroxidase assays of metal-loaded RbAng-1a and MmAng-2a (M = Cu, Co, or Ni) revealed no substantial activity relative to LPMOs, though the Cu-loaded forms showed weak oxidase activity (Fig. S6). These levels are comparable to those of free Cu²⁺ and known weak LPMO oxidases, suggesting that the primary function of Ang-1 and Ang-2 is metal binding or transport, though a modest enzymatic role cannot be fully excluded.”

3. Binding affinities:

The metal ion binding affinities for the proteins should be measured and compared to reported metal transport proteins and metalloenzymes. Binding kinetics may also provide insight into the function of these proteins. The thermal shift assays may be able to provide binding affinities (doi: 10.1038/s41598-018-37072-x).

We have now measured binding affinities as recommended by the reviewer. These are described in the manuscript as mentioned above, and the full results are reported in the Supplementary Information as figures S5, S11, S19 and S21.

4. Proposed function of Ang-1 and Ang-2:

While the proposal of an “anglerase” is an interesting idea, there is little experimental evidence provided to support this. While HupE was previously shown to be a Ni transporter, the comparison to the porin for glycan capture is not sufficient as the proposed mechanism is not similar between these two proteins. Not enough information about these anglerase-HupE/UreJ-2 proteins is presented. The proteins may be a part of homo- or hetero-multimeric complex where the flexible linker between the two domains may not allow for the proposed conformational change (AlphaFold-Multimer may give some insight on this possibility). More evidence needs to be provided or a stronger argument made before this proposed function should be accepted.

We understand the reviewer’s point of view here and thank them for their comments about the naming of anglerases. The basis of the anglerase naming arises from the predicted interdomain flexibility between the anglerase domain and the membrane-bound HupE/Ure-2 domain such that the anglerase can act to ‘fish’ for metal ions. We are not, unfortunately, in a position to provide biochemical experimental evidence of this interdomain flexibility. But, in response to the reviewer’s comments, we have now carried out an in-depth *in silico* analysis of the flexibility and dynamics between the anglerase and the attached HupE/Ure-2 domain, and also to the likelihood of the formation of multimeric complexes.

- 1) AlphaFold 3 Multimer shows a low score for dimers, trimers, tetramers and other tertiary conformations, suggesting no formation of homo-multimeric complexes (data not shown).
- 2) We have now emphasised in the manuscript that Alphafold 3 predicts *both* open and closed conformations of the Ang-HupE/UreJ-2 complexes, showing that the different conformations are thermodynamically reasonable.
- 3) In recognition that Alphafold 3 does not give direct information about protein dynamics, however, we have also subjected each of the Alphafold structures of the Ang-1- and Ang-2-HupE/UreJ-2 complexes to:
 - a) Protein flexibility analysis as described in <https://onlinelibrary.wiley.com/doi/full/10.1002/prot.26471>, and
 - b) protein motion and dynamics simulations using *normal mode analysis* in DynaMut2 <https://biosig.lab.uq.edu.au/dynamut2/>

We assessed the flexibility of the interdomain link using “N-H S² order analysis”. (A technique which is otherwise used to assess main chain disorder, and therefore, to infer protein flexibility, in interpreting solution-NMR protein structures. It does not report on interdomain dynamics.) The results for Ang-1- HupE/UreJ-2 are shown below where *a* and *b* indicate moderate disordered regions of the interdomain linker, which infers flexibility (the colour scale shows the predicted order, red = most disorder). Building on this finding, we then carried out a protein motion and dynamics analysis to assess interdomain dynamics, the results of which are shown below on the right. Here, the vector diagram (largest displacement shown as red arrows which predicts relative protein motion), unambiguously shows the high relative dynamics of the anglerase domain to that of the HupE/UreJ-2 domain. It also shows that the anglerase domain rotates away from the HupE/UreJ-2 domain and, in doing so, exposes the metal binding site to solution.

The results of the equivalent analysis for Ang-2 are shown below (for clarity, the protein dynamics are represented here as a superposition of structures rather than a vector diagram as for Ang-1). Here the interdomain link (denoted with arrow labelled *a*) is predicted to be similarly disordered. The results of the protein motion and dynamics analysis are less clear cut than that for Ang-1, as the AlphaFold structure of the Ang-2-HupE/Ure complex contains a large disordered alpha-helical loop (not shown) which dominates the relative sizes of predicted protein motion. Notwithstanding the complication, however, the Ang-2 domain, like Ang-1, shows significant predicted motion and dynamics in which the Ang-2 domain rotates against the HupE/UreJ-2 domain exposing the metal binding site to solution, in accord with the anglerase proposal.

The above analyses show the following:

- 1) the main chain of the interdomain link is predicted to be disordered and thereby flexible,
- 2) this flexibility then underpins the large interdomain dynamics between the anglerase domain and the membrane-bound HupE/UreJ-2 domain, which reveal that the anglerase domain can 'rotate' away from the mouth of the transmembrane pore to expose the metal binding site to solution,
- 3) The range of the relative positions of the anglerase domain with respect to the HupE/UreJ-2 domain is shown by the differing AlphaFold structures, which demonstrate both fully open and fully closed conformations.

- 4) We cannot find evidence from AlphaFold for the formation of homo-multimeric complexes.

Accordingly, while acknowledging the reviewer's well-made point about anglerases, we suggest that the naming is appropriate given the interdomain dynamics. In order to emphasise these dynamics, we have replaced the following text:

"The fact that AF3 models this conformation is not surprising given the lack of modelling of the surrounding solvent and membrane, which steers the model to maximising protein-protein interactions. The two domains, however, are linked by a disordered and presumably flexible region (176-184 for RbAng-1a and 175-180 for MmAng-2a), which gives the anglerase domains conformational flexibility with respect to the membrane-anchored HupE/UreJ-2. This flexibility allows the anglerase to adopt an open conformation where its N-terminus histidine residue is exposed to the surrounding milieu, from which it can sequester any adventitious transition metal ions."

By

“Using the validated AF3 model of Ang-1 (which aligns closely with its crystal structure), we modelled the full-length Ang-1– and Ang-2–HupE/UreJ-2 apparatus (Fig. 6). In both cases, the metal-binding domain faces the HupE/UreJ-2 domain in a compact “closed” conformation relative to the periplasm (Fig. 6a,b). This is consistent with the modelling context, which lacks solvent and membrane constraints, and thus steers the model to maximise protein-protein interactions favouring close domain interactions.”

And

“A flexible linker region (residues 176–184 in RbAng-1a, 175–180 in MmAng-2a) connects the anglerase and HupE/UreJ-2 domains. Flexibility analysis (Fig. S26) and normal mode simulations (Fig. S27) revealed significant conformational dynamics. These models suggest that the anglerase domain can rotate away from the channel “mouth” of the permease, exposing its N-terminal His residue to the surrounding milieu, from which it can sequester any adventitious transition metal ions (indeed, both conformations are modelled by AlphaFold 3, which further predicts low likelihoods of homo-multimeric complexes – data not shown).”

5. Minor issues:

On line 424 the pLDDT is reported to be ≤ 0.8 , which is a very poor pLDDT score. This is probably misreported metric.

We apologized for the mistake. The correct pLDDT is ≤ 80.0 . The data have been corrected in the new version of the manuscript.

Reviewer #4 (Remarks to the Author):

Again, thank you for your time and effort. We feel this paper is much improved by the comprehensive referees' comments and are grateful for your work.

Reviewer #5 (**RESPONSES IN GREEN**):

The manuscript entitled "Signal-strapping: a protein-sequence search method for the discovery of metalloproteins" reports the development of a method for the discovery of new metalloproteins containing a N-terminal coordinating residue, such as the histidine found in LPMOs. In particular, "signal strapping" consists in using the sequence of an identified signal peptide with appended a coordinating residue (e.g. histidine) for running a proteomic search. Mature proteins displaying such metal-binding residue at the N-terminal, after cleavage of the signal peptide, are successfully found. Four examples of newly identified metalloproteins have been expressed and characterized, thus demonstrating the applicability of the reported methodology.

In my opinion, the manuscript would be suitable for publication in Nature Communications after addressing some concerns. In particular, given that the "signal strapping" method is the primary focus of the manuscript, I believe that certain aspects of this method need to be described in greater detail. This would provide the reader with all the information required for

replicating the procedure, while better illustrating its potential applications.

Following are some suggestions that would help improving clarity and impact of the manuscript:

1) The first step of the signal strapping method is the choice of a known secreted protein, as described on page 3, lines 88-89: "I) take a single or consensus sequence of a known secreted protein from an organism (e.g. cellulase) and identify the SP using SignalP 6.0." It is not clear if there is a rationale behind the choice of the starting protein or it is simply a random selection. Could the authors provide an explanation for their choice?

We thank the reviewer for this question. Reviewer #1 had a similar comment. Our response is given above, but, for convenience, also repeated here below:

We have now included the following justifications.

DUF4198

"GH5 proteins are widely secreted across both eukaryotes and prokaryotes, offering broad taxonomic coverage in downstream searches."

DUF6702

There is already a justification at the start of this section, but we have now expanded it with the underlined text below.

*"Building on the discovery of Ni-containing DUF4198 proteins and their association with metal-dependent biochemistry, we selected the SP of the DUF4198-domain protein from *Hydrogenophaga* sp. IBVHS1 (Table 1) for signal strapping (Table 1). A blastp search (excluding *Hydrogenophaga* sp.) retrieved DUF4198 homologues and several uncharacterised proteins."*

Ang-1

*"We applied the signal strapping method using the signal peptide (SP) from a nickel superoxide dismutase (Pfam 09055, WP_230780104.1) of *Roseiconus lacunae* (Table 1). Nickel superoxide dismutases are known to coordinate Ni²⁺ via an N-terminal histidine."*

Ang-2

*"In a separate search, we used the SP of a GH16 family hemicellulase (IPR000757; SUPfam 49899; WP_230780104.1) from *Streptomyces* sp. YIM 130001 (Table 1). In contrast to the known metalloprotein SP used for Ang-1, GH16s are broadly secreted across bacteria and eukaryotes. A BLASTp search (excluding *Streptomyces* sp.) returned GH16 proteins, metalloproteinase inhibitors (I36), cellulases, and again, proteins with a C-terminal HupE/UreJ-2 domain, with N-terminal domains distinct from Ang-1 (hereafter Ang-2). Like Ang-1, Ang-2 displays a non-cytoplasmic domain after the SP, followed by the TM HupE/UreJ-2 at the C-terminal (Fig. S15a), and ConservFold results showed 100% conservation of His1 immediately following SP cleavage, along with highly conserved potential metal-coordinating amino acids in other parts of the sequences (Fig. S15b, Supplementary Data 4)."*

Further, given that the method aims at finding yet undiscovered metalloproteins with a N-terminal coordinating residue (histidine in the examples provided), why did not the authors try using a known LPMO protein as the starting point for their search?

Starting with a signal peptide from a known LPMO was much more likely to return known LPMO sequences and, for this initial paper using signal strapping, we wanted to exemplify the method's potential for discovering new (non-LPMO) metalloproteins. Subsequent work which will follow this paper will concentrate on using known LPMO signal peptides to discover new LPMO and LPMO-like proteins.

2) I understand that in step 3 of the method pipeline (BLAST and UNIPROT search using the "SP+HX" sequence as a query), a large number of sequences are found (since 500 is put as the maximum number). Can you give an idea of how many target sequences are found at this point, and if all of them are analysed in the subsequent steps?

Thank you. For each 500 sequences found in the blast search just one or two targeted sequences were selected and we run the subsequent steps for all the targets found. We added this info in the new version of the manuscript when applicable.

3) Step 4 of the method is the "manual" review of these sequences for finding a second coordinating residue. This is described at page 17-18, lines 583-585 (experimental section) as follows: "Once a novel amino acid sequence was found with a histidine as the putative N-terminal residue of a new protein, further analysis was conducted to search for additional amino acids capable of coordinating with metals, such as another histidine (as in a histidine brace motif) or aspartic/glutamic acid residues."

Are any specific criteria used for the selection? Which residues, other than histidine, are included?

Thank you for the comment. We look for additional amino acids capable of coordinating to a metal ion, e.g. histidine, cysteine or aspartic/glutamic acids. We have now made a comment to this effect in the manuscript (see below*).

Is the number of coordinating residues or their position in the sequence considered?

The number of other coordinate residues are always considered, if there is just the N-terminal histidine and no other coordinating residues in the protein, the sequences would be discarded.

The position also matters. If the second coordinating residue was found to come immediately after the n-terminal coordinating residue, the sequence was discarded since the metal coordination by both amino acid side chains is significantly hindered due to steric restraints. We have now clarified this point in the new version of the manuscript (see below*).

Are the sequences given a "score" and sorted?

Thank you for this very interesting idea. We do understand the attraction of scoring, especially if many new sequences are discovered by the method. However, since the typical number of plausible new sequences was 1 or 2 per search, we did not feel it necessary to rank them. In saying this, as we explore signal strapping more widely in the future, we will consider a score of merit as a useful means of ranking new sequences.

I expect that a sequence containing more than one internal histidine would have higher probability to have at least one of them involved in metal coordination with respect to a sequence containing just one glutamine. This aspect should be clarified.

This is an interesting comment and one surely that is true. It was not something we considered in our original analysis but accept that it's a valid notion. The following change* has been made to the material and methods section of the manuscript:

Replaced...

“Once a novel amino acid sequence was found with a histidine as the putative N-terminal residue of a new protein, further analysis was conducted to search for additional amino acids capable of coordinating with metals, such as another histidine (as in a histidine brace motif) or aspartic/glutamic acid residues.”

With...

“Once a novel amino acid sequence was found with a histidine as the putative N-terminal residue of a new protein, further analysis was conducted to search for additional amino acids capable of coordinating with metals, such as another histidine (as in a histidine brace motif) or other residues capable of coordinating a metal ion, where sequences which had several potential conserved amino acids that could coordinate given higher consideration. Any sequences which had the second amino acid in position 2 (i.e. immediately after the N-terminal histidine) were rejected as steric constraints would make simultaneous coordination by the side chains of both amino acids highly unlikely.”

4) Is there any correspondence between the identified internal coordinating residues (apart from the N-terminal His) and those actually found as metal ligands in the experimental structures? For example, Figure 5 shows that the metal binding site of RbAng-1a involves Asn41 as the internal ligand. Was this expected from the selection, or the sequence was selected because it contains other residues (e.g. histidine) that are not found at the metal binding site?

Albeit unusual, asparagine was included as a potential coordinating amino acid. We have now included this possibility in the revision* made immediately above.

Additional comments:

o The use of terms as "strapping" or "bootstrap" may not be readily clear to non-specialized audience. I believe that their meaning and context should be explained at their first appearance in the text.

Thank you. This is a useful comment. As such, we have made the following change to the manuscript.

Replaced..

“For ease of reference, we propose that the method is called signal strapping.”

with...

“For clarity, we refer to this method as “signal strapping”—a term derived from “bootstrapping,” reflecting the concept of initiating discovery using a minimal, adapted sequence.”

The "signal strapping procedure" is detailed in the introduction section, where also some results are shown regarding the occurrence of N-terminal amino acids in mature vs non-secreted proteins (as reported in figure 1, panels d-f). I believe that this part would be more appropriately placed at the beginning of results and discussion. Accordingly, I would split figure 1, leaving panels 1 a-c in the introduction and panels d-f as a separate figure in the results section.

Thank you for this suggestion. The reviewer makes a good point. Rather than splitting figure 1 as the reviewer suggests, we have now started the results and discussion section

earlier, now including the signal strapping method and the propensity analysis for N-terminus amino acids. We hope that this addresses the reviewer's point adequately.

o Figures in the supporting information should be all recalled in the main text and in sequential order.

Thank you. This has been attended to.

Reviewer #6 (Remarks to the Author):

Again, thank you. This is a great initiative by the journal and we thank you for your work.